# FocalPolicy: Frequency-Optimized Chunking and Locally Anchored Flow Matching for Coherent Visuomotor Policy

Qian He [* 1 2]  Zhenshuo Yang [* 1 2]  Wenqi Liang [3]  Chunhui Hao [1]  Nicu Sebe [3]  Jiandong Tian [1]

## Abstract

Visuomotor policies aim to learn complex manipulation tasks from expert demonstrations. However, generating smooth and coherent trajectories remains challenging, as it requires balancing proximal precision with distal foresight. Existing approaches typically focus on optimizing intra-chunk action distributions, often neglecting the inter-chunk coherence. Consequently, inter-chunk discontinuities significantly impede the learning of coherent long-horizon actions. To overcome this limitation and achieve a synergetic balance between precision and foresight, we propose FocalPolicy, a foresight-aware visuomotor policy that combines **F**requency-**O**ptimized **C**hunking with **L**ocally **A**nchored flow matching. We introduce a foresight composite objective that supervises time-domain alignment within the proximal actions while regularizing frequency-domain structure over multiple future action chunks to improve cross-chunk coherence. To efficiently learn complex action distributions, we design locally anchored sampling to enhance target signal propagation efficiency during consistency flow matching training. Extensive experiments demonstrate that FocalPolicy outperforms existing approaches and confirm the generalizability of our modules to other baselines. Project website: https://focalpolicy.github.io/.

## 1. Introduction

Recently, imitation learning-based visuomotor policies (Florence et al., 2021; Gong et al., 2025; Wang et al., 2026)

*Equal contribution [1]State Key Laboratory of Robotics and Intelligent Systems, Shenyang Institute of Automation, Chinese Academy of Sciences [2]University of the Chinese Academy of Sciences [3]University of Trento. Correspondence to: Jiandong Tian <tianjd@sia.cn>.

*Proceedings of the 43$^{rd}$ International Conference on Machine Learning*, Seoul, South Korea. PMLR 306, 2026. Copyright 2026 by the author(s).

have made significant progress in robotic manipulation. To mitigate compounding errors in imitation learning (Tu et al., 2021), action chunking has been adopted (Lai et al., 2022), allowing the policy to infer a multi-step action chunk per inference, rather than a single-step action (Zhao et al., 2023; Zhang et al., 2025b). Moreover, recent research emphasizes that accelerating the generation of action chunks is crucial for deploying visuomotor policies in the real world (Hu et al., 2024; Su et al., 2025c). Although effective, ensuring high-quality generation still necessitates a small number of iterative steps per inference, hindering real-world deployment. Consequently, efficiently generating action trajectories with minimal compounding errors remains a major bottleneck (Kim et al., 2025), which directly dictates the coherence and deployability of imitation learning policies (Park et al., 2025).

To address this issue, recent work (Song et al., 2023; Song & Dhariwal, 2024) aims to improve the accuracy of the trajectory in one-step generation. Prominent approaches achieve high-fidelity sampling through self-consistency constraints (Prasad et al., 2024; Lu et al., 2024) or consistency flow matching (Zhang et al., 2025a). Subsequent frameworks further improve performance by enhancing distributional fidelity (Wang et al., 2025c; Jia et al., 2024) or frequency-domain supervision (Su et al., 2025a; Zhong et al., 2025). As illustrated in Figure 1(a), these methods focus mainly on optimizing intra-chunk distributions, often neglecting inter-chunk discontinuities (Black et al., 2025), which leads to incoherence in subsequent action trajectories.

To mitigate the discontinuities between chunks, an intuitive approach is to increase the inference horizon. Nevertheless, directly fitting the more complex action distribution introduced by longer chunks leads to policy degradation. Therefore, this necessitates a foresight-aware policy that can simultaneously plan fine-grained immediate actions and coarse subsequent trajectories. We argue that *learning manipulation trajectories with a differentiated focus is crucial, as it enables the policy to not only finely align the fine-grained actions of proximal chunks but also attend to the coherence of subsequent chunks,* thereby bridging the inter-chunk discontinuities. To this end, we propose FocalPolicy, a foresight-aware visuomotor policy that

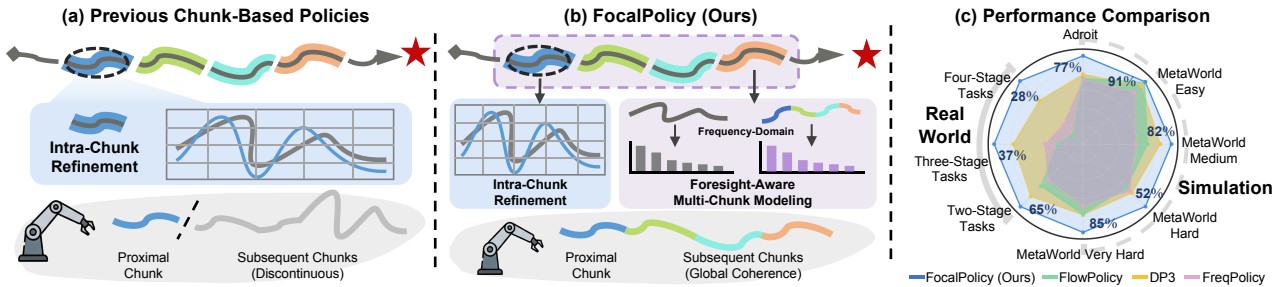

*Figure 1.* Comparison with chunk-based baselines. Unlike previous approaches (a) that prioritize intra-chunk refinement but overlook *inter-chunk discontinuities*, FocalPolicy (b) employs a *Foresight Composite Objective (FCO)* to synergize *proximal precision* with *distal coherence* across chunks. Extensive experiments (c) demonstrate that FocalPolicy outperforms state-of-the-art baselines.

combines Frequency-Optimized Chunking with Locally Anchored flow matching for the efficient generation of long-horizon actions, as illustrated in Figure 1(b). We introduce a **Foresight Composite Objective** that supervises time-domain alignment within the proximal actions while regularizing frequency-domain structure over multiple future action chunks to improve cross-chunk coherence. To overcome the complex action distributions within multi-chunk trajectories, we design **Locally Anchored Sampling** to enhance target signal propagation efficiency during training.

**Foresight Composite Objective (FCO).** While time-domain supervision enables precise local action alignment, frequency-domain features excel at capturing the global spatiotemporal relationships of action trajectories. Previous works (Su et al., 2025a; Zhong et al., 2025) have leveraged this synergy to significantly enhance the learning efficiency of single-chunk action distributions. However, the overlooked discontinuities in inter-chunk action distributions remain a major bottleneck for learning coherent manipulation trajectories. To bridge inter-chunk discontinuities, we propose the Foresight Composite Objective (FCO), which combines intra-chunk refinement in the proximal time-domain with foresight-aware multi-chunk modeling in frequency-domain, as illustrated in Figure 1. Unlike existing methods (Black et al., 2025; Park et al., 2025; Liu et al., 2025b) that regulate trajectory coherence by modifying the inference schemes of pre-trained models, FCO inherently strengthens the learning of action coherence at the fundamental level of policy training. As a generic and effective module, FCO can be seamlessly integrated into various action chunking-based policies.

**Locally Anchored Sampling (LAS).** Consistency flow matching enforces consistency constraints on adjacent time points randomly sampled during training, which often weakens the propagation efficiency of target signals. This represents a key bottleneck in improving training efficiency when modeling the complex action distributions of multi-chunk trajectories. To address this, we design Locally Anchored Sampling (LAS), which anchors one sampling time point to

a logit-normal distribution concentrated near the terminal time during training. This strategy reinforces direct supervision of the target distribution while preserving stochastic robustness, thereby significantly enhancing propagation efficiency and stabilizing training. Crucially, LAS acts as a general optimization booster that is broadly applicable to other consistency flow-based visuomotor policies.

We summarize the contributions of our work as follows:

- We propose FocalPolicy, a foresight-aware policy that explicitly addresses inter-chunk discontinuities in action generation via differentiated focus learning.

- We introduce the Foresight Composite Objective (FCO) to synergize proximal precision with distal coherence, serving as a versatile module applicable to various action chunking-based policies.

- We design Locally Anchored Sampling (LAS) to stabilize training for general consistency flow-based policies. Extensive experiments confirm the state-of-the-art performance of FocalPolicy and the effective generalizability of both FCO and LAS.

## 2. Related Work

### 2.1. Spatiotemporal Awareness for Visuomotor Policies

Understanding spatiotemporal relationships is crucial for robust robotic manipulation. Recent studies have enhanced this awareness through feature encoding and trajectory modeling (Liang et al., 2024; Ke et al., 2025; Wang et al., 2025a; Tian et al., 2025a; Lv et al., 2025; Su et al., 2025b; Liang et al., 2025; Wang et al., 2025b; Dong et al., 2026). At the representation level, DP4 (Liu et al., 2025c) incorporates a 3D Gaussian Splatting (3DGS) world model to guide the encoder in capturing spatiotemporal dynamics. Complementarily, recent work (Zhong et al., 2025) leverages frequency-domain analysis to structurally decouple global motion patterns from local details. Beyond feature-centric

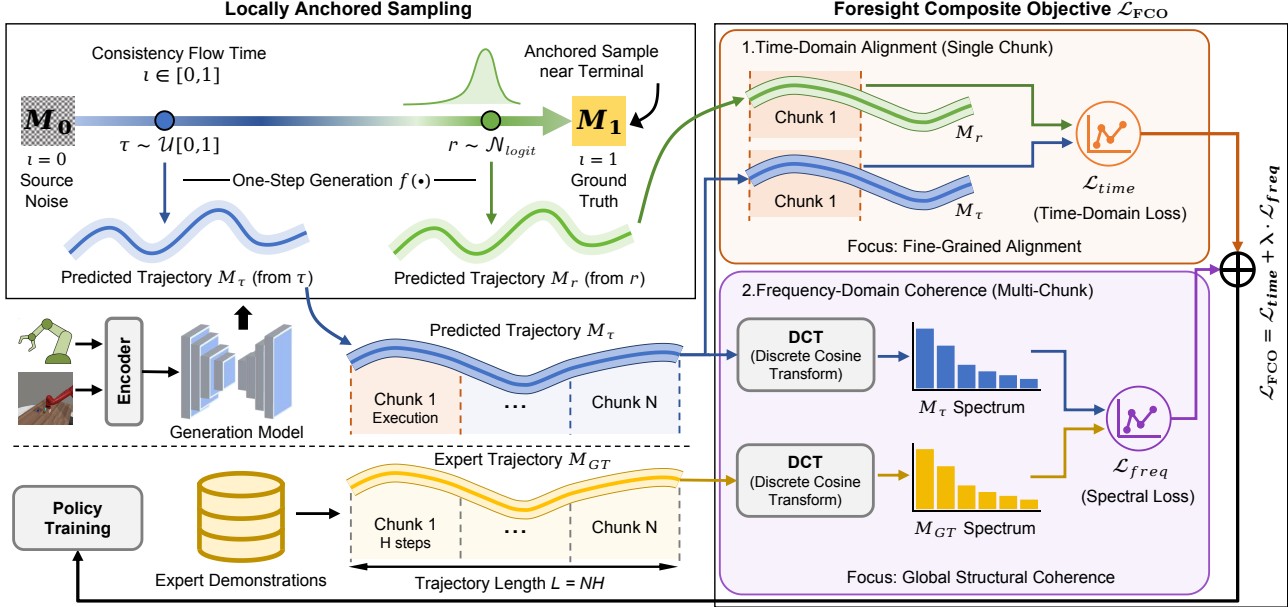

*Figure 2.* The pipeline of FocalPolicy. We propose *Locally Anchored Sampling (LAS)* to improve the training efficiency of consistency flow matching. The policy is optimized via a *Foresight Composite Objective (FCO)*, which synergizes proximal precision (via time-domain loss) with distal coherence (via frequency-domain loss) to generate smooth multi-chunk trajectories.

approaches, directly modeling object trajectories has proven effective (Wen et al., 2023; Yuan et al., 2025; Bharadhwaj et al., 2024; Xu et al., 2025). For instance, ORCA (Huey et al., 2025) employs a dense per-timestep reward to enable sequence-level spatiotemporal matching. Furthermore, spatiotemporal modeling facilitates inference acceleration; Dense Policy (Su et al., 2025c) proposes keyframe-based trajectory completion, significantly improving efficiency while maintaining manipulation performance.

### 2.2. Action Chunking-based Manipulation Policies

Action chunking (Zhao et al., 2023) mitigates compounding errors by predicting multi-step actions in a single pass, a paradigm now widely adopted in diffusion and flow-based policies. To ensure coherent real-time manipulation, RTC (Black et al., 2025) proposes an inference-time algorithm to overcome the discontinuities between action chunks. Recently, ManiCM (Lu et al., 2024) and FlowPolicy (Zhang et al., 2025a) have advanced one-step inference by utilizing consistency distillation and consistency flow matching (CFM) (Yang et al., 2024), respectively. Building on this, frequency consistency constraints are integrated into consistency flow policies (Su et al., 2025a), further strengthening the temporal correlation of action trajectories. While effective, the aforementioned approaches prioritize intra-chunk fidelity yet overlook inter-chunk discontinuities in long-horizon execution. To bridge this gap, we shift from single-chunk refinement to foresight-aware multi-chunk modeling, ensuring near-term precision and long-term coherence.

## 3. Preliminaries

Flow Matching (FM) (Lipman et al., 2023) models a $d$-dimensional data distribution by learning a time-dependent vector field $v_\theta(\mathbf{x}, \tau) : \mathbb{R}^d \times [0, 1] \to \mathbb{R}^d$, which defines an ODE that transports samples from the noise distribution $p_0$ (typically Gaussian) to the target distribution $p_1$ at $\tau = 1$. FM trains $v_\theta$ to match the conditional target field $u_\tau(\mathbf{x}_\tau \mid \mathbf{x}_1)$ given data samples $\mathbf{x}_1$. Under the Optimal Transport (OT) path (Amos et al., 2023) $\mathbf{x}_\tau = (1 - \tau)\mathbf{x}_0 + \tau\mathbf{x}_1$, the target field reduces to $u_\tau(\mathbf{x}_\tau \mid \mathbf{x}_1) = \mathbf{x}_1 - \mathbf{x}_0$. Accordingly, the FM objective is formulated as follows:

$$\mathcal{L}_{\text{FM}}(\theta) = \mathbb{E}_{\mathbf{x}, \tau} \|v_\theta(\mathbf{x}_\tau, \tau) - u_\tau(\mathbf{x}_\tau|\mathbf{x}_1)\|^2. \quad (1)$$

Although FM is effective, the iterative ODE integration required at inference time incurs a high number of function evaluations (NFE), hindering real-time deployment.

To enable one-step generation, Consistency Flow Matching (CFM) (Yang et al., 2024) enforces straight flows (Lee et al., 2023) via velocity consistency, $v_\theta(\mathbf{x}_\tau, \tau, c) \equiv v_\theta(\mathbf{x}_0, 0, c)$, where $c$ denotes the condition. Accordingly, CFM enforces this straight-line behavior by minimizing a self-consistency loss between predictions at neighboring flow times $\tau$ and $\tau + \Delta\tau$, resulting in the following training objective:

$$\mathcal{L}_{\text{CFM}}(\theta) = \mathbb{E}_{\mathbf{x}, \tau, c} \Big[ \|f_\theta(\mathbf{x}_\tau, \tau, c)$$
$$- f_{\theta^-}(\mathbf{x}_{\tau+\Delta\tau}, \tau + \Delta\tau, c)\|^2$$
$$+ \alpha \|v_\theta(\mathbf{x}_\tau, \tau, c) - v_{\theta^-}(\mathbf{x}_{\tau+\Delta\tau}, \tau + \Delta\tau, c)\|^2 \Big],$$
$$(2)$$

where $f_\theta(\mathbf{x}_\tau, \tau, c) = \mathbf{x}_\tau + (1-\tau)v_\theta(\mathbf{x}_\tau, \tau, c)$, $\theta^-$ denotes the network parameters updated via exponential moving average (EMA), and $\alpha > 0$ balances the loss terms. Under this constraint, CFM realizes straight-line noise-to-data transport, enabling one-step generation at inference time.

FlowPolicy (Zhang et al., 2025a) first introduces the consistency flow-based framework into robotic imitation learning to accelerate action inference. At each environment timestep $t$, the policy aims to generate an action chunk $\mathbf{A}_t = [\mathbf{a}_t, \mathbf{a}_{t+1}, \ldots, \mathbf{a}_{t+H-1}] \in \mathbb{R}^{H \times d}$, representing $H$ future $d$-dimensional actions. Accordingly, the flow state $\mathbf{x}_\tau$ in Eq. (2) is defined as the noisy action chunk $\mathbf{A}_\tau$ at flow time $\tau$, conditioned on the current observation $o_t$. The training objective is adapted as:

$$
\begin{aligned}
\mathcal{L}_{\mathrm{FP}}(\theta) = \mathbb{E}_{\mathbf{A}_\tau, \tau, o_t} \Big[ &\| f_\theta(\mathbf{A}_\tau, \tau, o_t) \\
&- f_{\theta^-}(\mathbf{A}_{\tau+\Delta\tau}, \tau + \Delta\tau, o_t) \|^2 \\
&+ \alpha \| v_\theta(\mathbf{A}_\tau, \tau, o_t) - v_{\theta^-}(\mathbf{A}_{\tau+\Delta\tau}, \tau + \Delta\tau, o_t) \|^2 \Big].
\end{aligned}
\tag{3}
$$

As shown in Eq. (3), it relies solely on self-consistency between adjacent time steps within an action chunk. This supervision lacks explicit guidance from the expert trajectory, making it difficult to perceive long-horizon motion trends and model complex multi-chunk action distributions.

## 4. Methodology

In this work, we propose **FocalPolicy**, a foresight-aware visuomotor policy as illustrated in Figure 2. Unlike prior works that focus on time-domain optimization within a single action chunk, FocalPolicy introduces a **Foresight Composite Objective (FCO)**, an optimization target that integrates proximal time-domain alignment with multi-chunk frequency-domain structural regularization (Sec. 4.1). Furthermore, to enhance the target-signal propagation efficiency during training, we design a **Locally Anchored Sampling (LAS)** strategy (Sec. 4.2).

### 4.1. Foresight Composite Objective

To achieve seamless long-horizon execution, we identify prioritized learning as a key necessity: the policy should ensure fine-grained fidelity for proximal actions while maintaining coarse structural coherence for distal trajectories. We therefore propose the Foresight Composite Objective (FCO). Specifically, we explicitly concatenate $N$ consecutive expert action chunks into a macro-trajectory of horizon length $L = NH$, denoted as $\mathbf{M}_t \in \mathbb{R}^{L \times d}$:

$$
\mathbf{M}_t = \mathrm{Concat}\big( \mathbf{A}_t, \mathbf{A}_{t+H}, \ldots, \mathbf{A}_{t+(N-1)H} \big). \tag{4}
$$

Moreover, prior work (Zhang et al., 2025a) indicates that low-frequency components encode global motion trends,

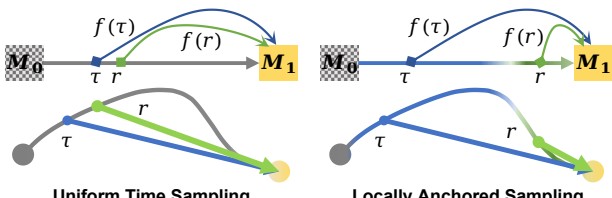

*Figure 3.* Time sampling comparison. Left: Standard uniform sampling draws $(\tau, r)$ uniformly along the flow trajectory, which can attenuate target-signal propagation for early $\tau$. Right: Our locally anchored sampling biases $r$ toward the terminal region to strengthen target-signal propagation.

while high-frequency components represent fine-grained details. These properties align well with our foresight requirement. Motivated by this observation, we apply the Discrete Cosine Transform (DCT) (Khayam, 2003) to project macro-trajectories into the frequency domain. For each action dimension $d$, the DCT coefficients $\mathcal{C}_d \in \mathbb{R}^L$ are computed as follows:

$$
\mathcal{C}_d[u] = c_u \sum_{i=0}^{L-1} \mathbf{M}_t[i, d] \cos\left( \frac{\pi}{L}(i + 0.5)u \right), \tag{5}
$$

where $u = 0, \ldots, L - 1$ is the frequency index, and $c_u$ are normalization constants with $c_0 = \sqrt{1/L}$ and $c_u = \sqrt{2/L}$ for $u > 0$. Eq. (5) further shows that each DCT coefficient aggregates information across all timesteps, providing a global view of the macro-trajectory.

To model both distal motion trends and proximal execution details, FCO combines intra-chunk refinement in the time domain with foresight-aware multi-chunk modeling in the frequency domain. Concretely, we first define the multi-chunk frequency-domain loss $\mathcal{L}_{\mathrm{freq}}$ as the squared $L_2$ distance between the DCT coefficients of the predicted trajectory $\hat{\mathbf{M}}_t$ and the expert trajectory $\mathbf{M}_t$:

$$
\mathcal{L}_{\mathrm{freq}} = \mathbb{E} \left\| \mathrm{DCT}(\hat{\mathbf{M}}_t) - \mathrm{DCT}(\mathbf{M}_t) \right\|^2, \tag{6}
$$

where $\mathrm{DCT}(\cdot)$ denotes the DCT operator defined in Eq. (5). The final Foresight Composite Objective, denoted as $\mathcal{L}_{\mathrm{FCO}}$, is then formulated as a weighted combination of the proximal single-chunk time-domain consistency loss $\mathcal{L}_{\mathrm{time}}$ (Eq. (14)) and the multi-chunk frequency-domain loss $\mathcal{L}_{\mathrm{freq}}$:

$$
\mathcal{L}_{\mathrm{FCO}} = \mathcal{L}_{\mathrm{time}} + \lambda \cdot \mathcal{L}_{\mathrm{freq}}. \tag{7}
$$

Here, $\lambda = 10^{-4}$ is a balancing coefficient chosen to equilibrate the relative scales of the time-domain and frequency-domain loss components. This hybrid objective supervises time-domain alignment within proximal actions while regularizing frequency-domain structure over multiple future action chunks to improve cross-chunk coherence.

Crucially, although $\mathcal{L}_{\mathrm{freq}}$ is defined in the frequency domain, Parseval's theorem (Rao & Yip, 2014) ensures that

its error signal is consistent with the time-domain $\ell_2$ objective. In FCO, $\mathcal{L}_{\text{time}}$ supervises only a single chunk of length $H$, while $\mathcal{L}_{\text{freq}}$ supervises multiple chunks spanning a length of $L$, and the complete task trajectory is significantly longer than $L$. Under this formulation, the resulting network-parameter gradients remain non-conflicting:

$$\left\langle \nabla_\theta \mathcal{L}_{\text{freq}}, \ \nabla_\theta \mathcal{L}_{\text{time}}^{(H)} \right\rangle \ > \ 0. \tag{8}$$

Thus, optimizing $\mathcal{L}_{\text{freq}}$ does not introduce conflicting optimization directions. A detailed derivation and further discussion on spectral sensitivity, along with the complete training process of FocalPolicy, are provided in the Appendix A.

### 4.2. Locally Anchored Sampling

While FCO effectively strengthens global coherence, it requires modeling the macro-trajectory, which greatly increases distributional complexity. As discussed in Sec. 3, FlowPolicy imposes consistency constraints between adjacent random time points. However, this often weakens the target-signal propagation efficiency, hindering the learning of complex distributions.

**Definition 4.1.** *To quantify the effectiveness of information flow during training, we define the Target-signal Propagation Efficiency $\mathcal{E}(\tau)$ as the fidelity of the consistency gradient $g_{cons}(\tau, r)$ relative to the ideal supervised gradient $g_{sup}(\tau)$:*

$$\mathcal{E}(\tau) := -\mathbb{E}_{r \mid \tau}\left[\|g_{\text{cons}}(\tau, r) - g_{\text{sup}}(\tau)\|^2\right]. \tag{9}$$

**Remark 4.1.** Under standard uniform sampling, the high variance of intermediate targets causes $\mathcal{E}(\tau)$ to decay rapidly as $\tau$ moves away from the terminal state, leading to suboptimal target distribution learning.

To improve the target-signal propagation efficiency, we propose Locally Anchored Sampling (LAS). Figure 3 illustrates the difference between standard uniform sampling and our locally anchored strategy, which biases the anchor time toward the terminal region. Specifically, we sample a flow time $\tau \sim \mathcal{U}[0, 1]$ and an anchor time $r$ from a logit-normal distribution. We formulate $r$ as:

$$r = \text{sigmoid}(\mu_r + \sigma_r \epsilon), \quad \epsilon \sim \mathcal{N}(0, 1), \tag{10}$$

where $\mu_r, \sigma_r$ control the anchor bias. By biasing $r$ toward the terminal boundary, we effectively strengthen the target-signal propagation efficiency in consistency training. However, rather than fixing $r$ to the terminal time, we keep it stochastic to stabilize training and preserve temporal continuity along the flow. In practice, we set $\mu = 4.0$ and $\sigma = 1.6$, mapping to a median time of $r \approx 0.982$ with a heavy left tail, which effectively provides the desired near-terminal supervision.

Following the Optimal Transport (OT) construction, we define the interpolated states at times $\tau$ and $r$ as follows:

$$\mathbf{M}_\tau = (1 - \tau)\mathbf{M}_0 + \tau\mathbf{M}_1, \tag{11}$$
$$\mathbf{M}_r = (1 - r)\mathbf{M}_0 + r\mathbf{M}_1, \tag{12}$$

where $\mathbf{M}_0$ is a Gaussian noise trajectory, and $\mathbf{M}_1$ corresponds to the target expert macro-trajectory $\mathbf{M}_t$. Conditioned on observation $o_t$, the model predicts velocity fields $v_\theta(\mathbf{M}_\tau, \tau, o_t)$ and $v_\theta(\mathbf{M}_r, r, o_t)$. We define the model predictions of the target action trajectory from any given flow state as follows:

$$f_\theta(\mathbf{M}_{\tau'}, \tau', o_t) = \mathbf{M}_{\tau'} + (1 - \tau')v_\theta(\mathbf{M}_{\tau'}, \tau', o_t), \tag{13}$$

where $\tau' \in \{\tau, r\}$. To enforce the straight-flow property, we impose a consistency constraint between the predictions at the uniform point $\tau$ and the anchored point $r$. The time-domain consistency loss is defined as:

$$\mathcal{L}_{\text{time}} = \mathbb{E}_{\tau, r}\left[\|[f_\theta(\mathbf{M}_\tau, \tau, o_t) - f_{\theta^-}(\mathbf{M}_r, r, o_t)]_{1:H}\|^2\right], \tag{14}$$

where the subscript $1 : H$ indicates that the consistency loss is computed exclusively on the first $H$ timesteps (*i.e.*, the proximal single chunk) of the predicted macro-trajectory. By assuming monotonically decreasing error near the terminal boundary, we mathematically demonstrate that our locally anchored sampling strategy achieves a strictly superior propagation efficiency $\mathcal{E}(\tau)$ compared to the standard random sampling strategy. Detailed theoretical derivations and proofs are provided in Appendix B.

## 5. Experiments

This section presents a comprehensive evaluation of FocalPolicy. We first detail the setup, baselines, and metric. Next, we demonstrate the superiority of FocalPolicy and validate the generalizability and distinct contributions of FCO and LAS. Finally, we conduct extensive ablation studies on 24 tasks to verify the effectiveness of our design choices.

### 5.1. Experiments Setup

**Simulation experiments.** To achieve a comprehensive evaluation across diverse robotic manipulation skills, we select 53 tasks from the Adroit (Rajeswaran et al., 2017) and Meta-World (Yu et al., 2020) simulation platforms. The Adroit is primarily designed for evaluating dexterous hand manipulation tasks, while MetaWorld focuses on parallel gripper manipulation tasks. All tasks were constructed using simulators such as MuJoCo (Todorov et al., 2012) and SaPien (Xiang et al., 2020). For expert demonstration data, comprising 10 trajectories of 200 steps per task, we employed well-trained heuristic policies to ensure the generation of high-quality datasets, with Adroit using the VRL3 (Wang et al., 2022)

*Table 1.* Main results in simulation. We report the number of function evaluations (NFE) and success rates (%) for each method across benchmarks. Averages computed over **53** tasks. * indicates results reproduced using the same expert demonstrations for fair comparison.

| Method \ Task | NFE | Adroit (3) | MetaWorld Easy (28) | MetaWorld Medium (11) | MetaWorld Hard (6) | MetaWorld Very Hard (5) | Average |
|---|---|---|---|---|---|---|---|
| DP | 10 | 31.7 | 83.6 | 31.1 | 9.0 | 26.6 | 55.9 |
| MainCM | 1 | 72.3 | 83.6 | 55.6 | 33.3 | 67.0 | 69.9 |
| SDM | 1 | 74.0 | 86.5 | 65.8 | 35.8 | 71.6 | 74.4 |
| DP3* | 10 | $70.4 \pm 3.2$ | $89.3 \pm 0.1$ | $77.2 \pm 2.7$ | $47.1 \pm 2.3$ | $78.1 \pm 0.9$ | 79.9 |
| FlowPolicy* | 1 | $69.0 \pm 2.6$ | $91.1 \pm 0.2$ | $71.9 \pm 1.2$ | $45.2 \pm 0.9$ | $77.9 \pm 0.6$ | 79.4 |
| FreqPolicy* | - | $68.6 \pm 0.8$ | $85.4 \pm 0.2$ | $66.6 \pm 2.0$ | $46.4 \pm 2.1$ | $74.4 \pm 2.4$ | 75.1 |
| **Ours** | 1 | $\mathbf{76.9 \pm 2.2}$ | $\mathbf{91.4 \pm 0.1}$ | $\mathbf{81.9 \pm 1.6}$ | $\mathbf{51.8 \pm 3.8}$ | $\mathbf{85.1 \pm 1.6}$ | **83.6** |

*Table 2.* Comparing FocalPolicy with more baselines on 7 simulation tasks. We compare Mamba Policy, DP3, FlowPolicy, and their variants, termed DP3 w. FCO and FlowPolicy w. LAS.

| Method \ Task | Adroit | | | MetaWorld | | | | Average |
|---|---|---|---|---|---|---|---|---|
| | Hammer | Door | Pen | Sweep-Into | Hand-Insert | Pick-Place | Disassemble | |
| Mamba Policy* | $99.7 \pm 0.6$ | $59.0 \pm 2.0$ | $58.3 \pm 3.8$ | $59.3 \pm 17.0$ | $15.3 \pm 4.5$ | $0.0 \pm 0.0$ | $79.7 \pm 4.7$ | 53.1 |
| FreqPolicy* | $94.3 \pm 2.1$ | $58.3 \pm 3.8$ | $53.0 \pm 3.5$ | $19.3 \pm 8.5$ | $16.0 \pm 3.0$ | $57.7 \pm 4.2$ | $85.3 \pm 1.5$ | 54.9 |
| **Ours** | $\mathbf{100.0 \pm 0.0}$ | $\mathbf{63.2 \pm 5.3}$ | $\mathbf{67.5 \pm 1.3}$ | $\mathbf{70.7 \pm 22.0}$ | $\mathbf{29.3 \pm 16.2}$ | $70.7 \pm 4.2$ | $\mathbf{86.7 \pm 1.5}$ | **69.7** |
| DP3* | $98.3 \pm 2.9$ | $53.5 \pm 5.1$ | $59.3 \pm 3.5$ | $39.3 \pm 25.3$ | $14.3 \pm 2.9$ | $60.7 \pm 11.9$ | $79.3 \pm 7.0$ | 57.8 |
| DP3 w. Ours | $98.3 \pm 1.5$ | $58.3 \pm 2.5$ | $59.7 \pm 2.1$ | $47.0 \pm 24.3$ | $15.3 \pm 0.6$ | $61.7 \pm 9.3$ | $84.0 \pm 5.6$ | 60.6 |
| FlowPolicy* | $94.0 \pm 0.0$ | $53.0 \pm 6.2$ | $61.0 \pm 3.5$ | $37.3 \pm 31.8$ | $16.0 \pm 2.6$ | $55.3 \pm 2.3$ | $71.0 \pm 7.8$ | 55.4 |
| FlowPolicy w. Ours | $98.7 \pm 2.3$ | $61.3 \pm 3.2$ | $62.7 \pm 4.5$ | $62.3 \pm 17.6$ | $16.5 \pm 0.7$ | $66.3 \pm 4.6$ | $75.0 \pm 6.0$ | 63.3 |

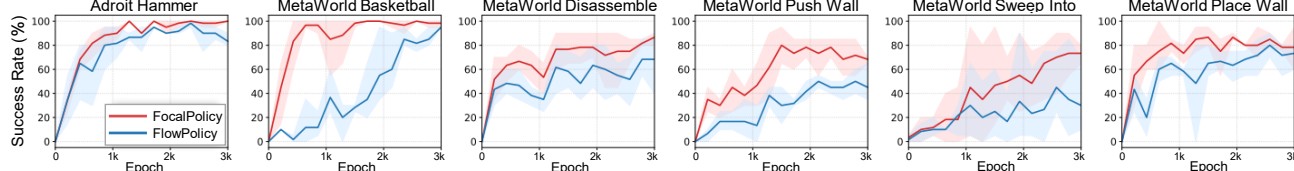

*Figure 4.* Learning efficiency. We report learning curves of FocalPolicy and FlowPolicy across six tasks. Benefiting from higher target signal propagation efficiency during training, FocalPolicy converges faster to higher success rates than FlowPolicy.

algorithm and MetaWorld employing the PPO (Schulman et al., 2017) algorithm.

**Real-world experiments.** Referencing the multi-stage task design in (Liu et al., 2025a; Tian et al., 2025b), we curate six long-horizon tasks to evaluate the real-world performance of FocalPolicy. We categorize these tasks into three levels, ranging from two to four stages: Two-stage tasks: 1) **Water Pouring**. The robot pours water from a kettle into a cup and relocates the kettle. 2) **Drawer Loading**. The robot places an object into an open drawer and pushes it closed. Three-stage tasks: 3) **Pot Loading**. The robot removes a pot lid, adds an ingredient, and recovers the pot. 4) **Tower Stacking**. The robot sequentially stacks three objects of varying shapes. Four-stage tasks: 5) **Cup Matching**. The robot relocates three cups onto corresponding saucers. 6) **Object Sorting**. The robot sorts four items into two bins.

Figure 5 illustrates the setup and stages for representative tasks, with details in Appendix C.4.

**Baselines and Implementation Details.** We compare FocalPolicy against state-of-the-art visuomotor policies designed for 3D point-cloud observations and action chunking. Specifically, we re-evaluate **FlowPolicy** (Zhang et al., 2025a), **DP3** (Ze et al., 2024), **Mamba Policy** (Cao et al., 2025) and **FreqPolicy** (Zhong et al., 2025) on our datasets as primary baselines. For **DP** (Chi et al., 2023), **ManiCM** (Lu et al., 2024), and **SDM** (Jia et al., 2024), we include their reported metrics for comparison. All baselines utilize original configurations, while for the parameters specific to FocalPolicy, we set the action chunk size $H = 4$ and the number of aggregated chunks $N = 3$. We adopt a consistent training and evaluation setup across all methods, with details provided in Appendix C.1.

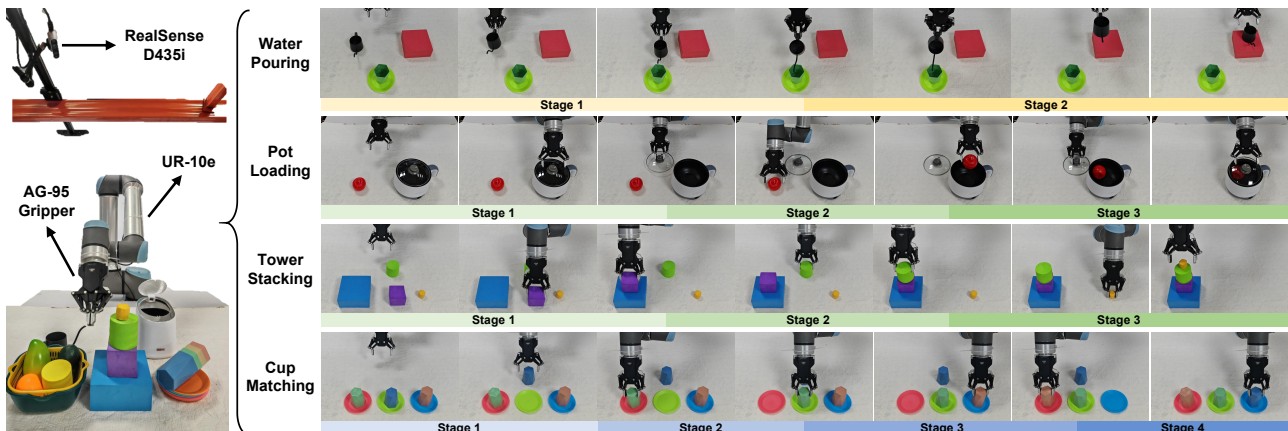

*Figure 5.* Real-world experimental setup and stage definitions. Left: Real-world experimental setup utilizing a UR-10e robotic arm equipped with an AG-95 gripper and a RealSense D435i camera. Right: Illustration of stage-wise definitions for four representative tasks.

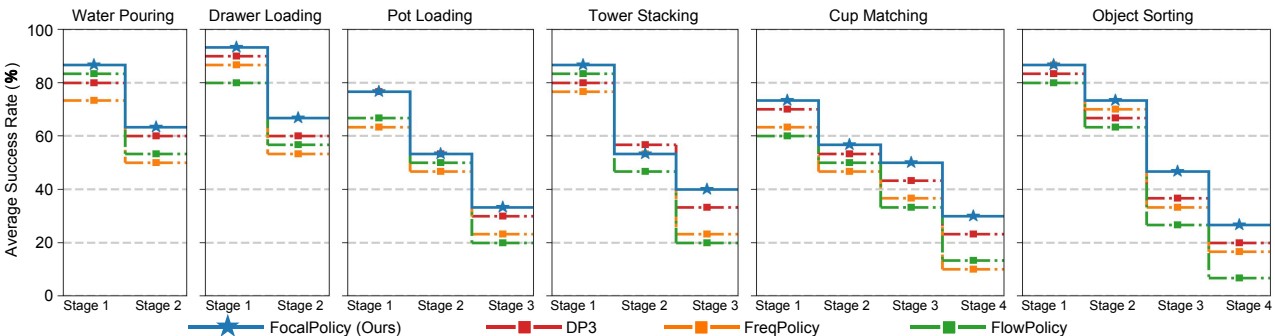

*Figure 6.* Real-world main results. We evaluate FocalPolicy, FlowPolicy, DP3, and FreqPolicy on six tasks. We report the average success score at each stage, and the horizontal segments indicate the policy performance across different stages of each task.

**Evaluation Metric.** Following the evaluation protocol in DP3, we evaluate each task across three runs using seeds 0, 1, and 2. For each seed, we evaluate the policy 20 times every 200 epochs and report the average of the top five success rates, along with the mean and variance across seeds.

### 5.2. Simulation Results

**1) Main Results.** Table 1 summarizes the results on 53 tasks, where FocalPolicy reaches an average success rate of 83.6%, outperforming DP3 (79.9%), FlowPolicy (79.4%), and FreqPolicy (75.1%). Table 2 further compares FocalPolicy with more baselines on 7 representative tasks. FocalPolicy achieves the highest success rate on all tasks, with an average of 69.7%. These results demonstrate the effectiveness of foresight-aware multi-chunk modeling.

**2) Module Generalizability.**

To evaluate the generalizability of FCO and LAS, we directly integrate these modules into baseline models, denoted as DP3 w. Ours and FlowPolicy w. Ours. As shown in Table 2, DP3 w. Ours (60.6%) improves the average success rate by 2.8% over DP3, while FlowPolicy w. Ours

(63.3%) surpasses FlowPolicy by 7.9%. These improvements demonstrate that the benefits of FCO and LAS are not confined to FocalPolicy but are generalizable to similar model architectures.

**3) Mitigation of Compounding Errors and Discontinuities.** As shown in Figure 7, we visualize the predicted spatial trajectories alongside the corresponding expert trajectories, and compute their Euclidean distances to the expert ground truth to evaluate the compounding errors. Under the same 12-step prediction horizon, FocalPolicy tracks the expert trajectories more closely and exhibits significantly lower compounding errors compared to FlowPolicy.

To rigorously quantify the inter-chunk discontinuities, we further adopt Action Total Variation (ATV) (Park et al., 2025) to measure the temporal coherence of the generated action sequences. The ATV is defined as:

$$\text{ATV} = \frac{1}{(L-1) \cdot d} \sum_{t=1}^{L-1} \sum_{j=1}^{d} |a_{t+1}^j - a_t^j|, \quad (15)$$

where $t$ denotes the time step in a trajectory of length $L$, and $j$ denotes the action dimension up to $d$. As reported

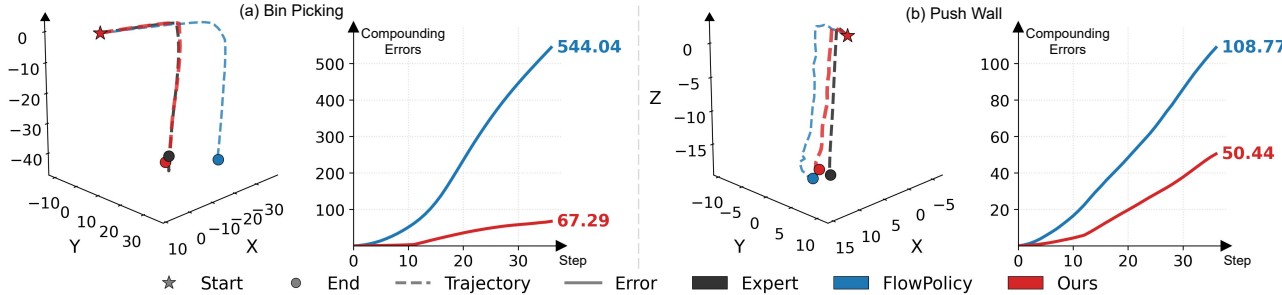

*Figure 7.* Comparison of compounding errors. We visualize the 3D end-effector trajectories (left) inferred by FocalPolicy and FlowPolicy on two representative simulation tasks (*Bin Picking* and *Push Wall*), given the same initial state. The corresponding error curves (right) represent the *Euclidean distance* between the predicted and ground-truth coordinates ($||\mathbf{p}_{\text{pred}} - \mathbf{p}_{\text{gt}}||_2$) at each time step.

*Table 3.* Quantitative comparison of Action Total Variation (ATV ↓). Lower values indicate higher temporal coherence and smoother action sequences across chunks.

| Task | FlowPolicy | **Ours** |
|---|---|---|
| Bin Picking | 0.99 | **0.87** |
| Disassemble | 0.35 | **0.21** |
| Pick Out of Hole | 0.94 | **0.92** |
| Pick Place | **0.03** | 0.03 |
| Pick Place Wall | 0.42 | **0.39** |
| Push Wall | 0.39 | **0.33** |
| Shelf Place | 0.48 | **0.46** |
| Sweep Into | 1.66 | **1.45** |

*Table 4.* Ablation study of component effectiveness on 24 tasks. We report the average success rate (%) across tasks.

| Designs | A (3) | M (11) | H (5) | VH (5) | **Average** |
|---|---|---|---|---|---|
| **Ours** | **76.9 ± 2.2** | **81.9 ± 1.6** | **62.2 ± 4.6** | **85.1 ± 1.6** | **77.9** |
| w/o LAS | 71.7 ± 1.8 | 65.6 ± 2.9 | 47.8 ± 1.8 | 73.3 ± 1.1 | 64.3 |
| w/o FCO&LAS | 46.2 ± 12.1 | 56.5 ± 3.2 | 43.1 ± 0.2 | 64.7 ± 2.4 | 54.1 |
| $\mathcal{L}_{\text{time}}$-Only FCO | 34.3 ± 1.2 | 65.9 ± 4.8 | 49.4 ± 2.1 | 78.1 ± 1.7 | 61.1 |
| $\mathcal{L}_{\text{freq}}$-Only FCO | 74.0 ± 1.0 | 74.4 ± 1.9 | 60.1 ± 1.3 | 81.4 ± 1.8 | 72.8 |
| Fixed-$r$ LAS | 70.4 ± 2.2 | 77.4 ± 2.0 | 59.0 ± 3.7 | 80.2 ± 1.7 | 73.3 |

*Table 5.* Horizon Ablation. We evaluate the impact of chunk size $H$ and the number of aggregated chunks $N$ across 24 tasks. We report the success rates under different settings of $H$ and $N$.

| $H$ | $N$ | A (3) | M (11) | H (5) | VH (5) | **Average** |
|---|---|---|---|---|---|---|
| | 2 | 73.6 ± 3.2 | 74.7 ± 1.1 | 59.1 ± 5.0 | 82.3 ± 2.1 | 72.9 |
| 4 | 3 | **76.9 ± 2.2** | **81.9 ± 1.6** | **62.2 ± 4.6** | **85.1 ± 1.6** | **77.9** |
| | 4 | 76.1 ± 3.4 | 76.5 ± 3.1 | 58.7 ± 1.8 | 82.1 ± 3.1 | 73.9 |
| | 5 | 70.4 ± 2.8 | 65.9 ± 3.5 | 49.5 ± 1.5 | 73.4 ± 2.1 | 64.6 |
| | 2 | 73.7 ± 2.3 | 78.5 ± 2.1 | 59.4 ± 1.7 | 80.3 ± 1.0 | 74.3 |
| 8 | 3 | 74.4 ± 1.8 | 71.2 ± 4.6 | 57.1 ± 0.6 | 78.9 ± 2.1 | 70.3 |
| | 4 | 73.4 ± 3.7 | 77.5 ± 0.8 | 60.0 ± 1.0 | 82.0 ± 1.4 | 74.3 |
| | 5 | 72.6 ± 3.1 | 75.4 ± 2.0 | 59.7 ± 2.3 | 80.0 ± 1.7 | 72.7 |
| | 2 | 72.7 ± 2.1 | 73.9 ± 1.3 | 61.4 ± 0.3 | 79.3 ± 2.2 | 72.3 |
| 12 | 3 | 73.6 ± 2.2 | 76.7 ± 0.7 | 61.6 ± 2.6 | 79.3 ± 1.7 | 73.7 |
| | 4 | 69.7 ± 2.9 | 72.9 ± 1.6 | 59.8 ± 2.2 | 77.3 ± 2.0 | 70.7 |
| | 5 | 69.4 ± 1.2 | 71.8 ± 1.8 | 61.3 ± 1.6 | 76.3 ± 1.7 | 70.3 |

in Table 3, FocalPolicy achieves consistently lower ATV scores than FlowPolicy across the evaluated tasks.

Together with the qualitative reduction in compounding errors, these quantitative results strongly demonstrate that FCO effectively mitigates the discontinuities between action chunks, thereby enabling FocalPolicy to maintain higher temporal coherence in long-horizon trajectory prediction.

**4) Learning Efficiency.** We track the success rates of FocalPolicy and FlowPolicy during training in Figure 4. The results show that FocalPolicy converges faster to a higher success rate than FlowPolicy, demonstrating superior learning efficiency. We attribute this improvement to LAS, which enhances target-signal propagation efficiency during consistency flow training.

For detailed implementation specifics and more results, please refer to the Appendix C.3.

### 5.3. Real World Results.

Figure 6 illustrates the real-world performance comparison of four methods: FocalPolicy, DP3, FreqPolicy, and FlowPolicy. FocalPolicy consistently achieves superior average success rates compared to other baselines, demonstrating its exceptional capability in capturing the complex distri-

butions of long-horizon actions. Notably, this performance advantage becomes most pronounced in the final stage. We attribute this success to the foresight composite objective, which guides the policy to bridge inter-chunk discontinuities, thereby mitigating compounding errors in imitation learning. Furthermore, FlowPolicy suffers a sharp performance decline across task stages, likely due to limited long-horizon modeling, whereas FocalPolicy mitigates this degradation by accurately predicting longer action chunks.

### 5.4. Ablation Study

To provide a solid validation of our proposed component effectiveness, we conduct extensive ablation studies across 24 tasks. The evaluation benchmarks encompass Adroit

(A) and MetaWorld tasks spanning three difficulty levels: Medium (M), Hard (H), and Very Hard (VH). Our analysis focuses on: **Component Effectiveness** and the **Impact of Horizon Hyperparameters**. To ensure a fair comparison, we conduct all experiments across three random seeds $(0, 1, 2)$ using the identical set of expert demonstrations.

**Component Effectiveness.** Table 4 presents the ablation results for the individual components of FocalPolicy. We first conduct ablation studies on FCO and LAS. Specifically, compared to FocalPolicy, the average success rates of variants 1) w/o LAS and 2) w/o FCO&LAS decrease by $13.6\%$ and $23.8\%$, respectively. To further validate specific design choices, we define additional variants: 3) $\mathcal{L}_{\text{time}}$-Only FCO: Replaces long-horizon frequency supervision with a long-horizon time-domain loss, validating the superiority of frequency-domain supervision in FCO; 4) $\mathcal{L}_{\text{freq}}$-Only FCO: Removes the time-domain loss from FCO to evaluate frequency-only supervision; and 5) Fixed-$r$ LAS: Restricts the sampling time $r$ in LAS strictly to the endpoint ($r = 1$). The superior performance of FocalPolicy demonstrates the benefits of FCO and LAS for visuomotor policies.

**Impact of Horizon Hyperparameters.** We investigate the model sensitivity to the action chunk size $H \in \{4, 8, 12\}$ and the number of aggregated chunks $N \in \{2, 3, 4, 5\}$. As shown in Table 5, $H = 4$ and $N = 3$ yields optimal performance. We attribute this to two factors: 1) Responsiveness vs. Stability ($H$): FocalPolicy's frequency regularization allows for shorter chunks ($H = 4$), facilitating rapid replanning while effectively mitigating the boundary discontinuities typically associated with short horizons; 2) Effective Spectral Context ($N$): $N = 3$ provides sufficient foresight to capture low-frequency trends without inducing the oversmoothing or latency observed at larger values (e.g., $N = 5$). Comprehensive ablation results are detailed in Appendix D.

## 6. Conclusion & Limitation

We introduce FocalPolicy, a foresight-aware visuomotor policy that mitigates inter-chunk discontinuities in action generation by employing coarse-to-fine prioritized planning. By leveraging a foresight composite objective and locally anchored sampling, it achieves higher success rates than recent methods. However, it does not incorporate foresight reasoning for high-dimensional decision-making. Furthermore, it does not explicitly address capability enhancements for handling suboptimal demonstrations, distribution shifts, or out-of-distribution generalization. Despite these challenges, we believe FocalPolicy serves as an important step towards achieving foresight-aware robotic manipulation.

## Acknowledgement

We thank all reviewers for their valuable suggestions. This work is supported by the Nation Key R&D Program of China under Grant 2024YFE0115500, the LiaoNing Revitalization Talents Program under Grant XLYC 2502011, the FIS project GUIDANCE (No. FIS2023-03251) and the EU Horizon project ELLIOT (No. 101214398).

## Impact Statement

This paper presents work whose goal is to advance the field of Machine Learning. There are many potential societal consequences of our work, none of which we feel must be specifically highlighted here.

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

# Appendix

## Appendix Contents

---

## A. Theoretical Analysis: Gradient Consistency of Foresight Composite Objective

In this appendix, we analyze the spectral component in the *Foresight Composite Objective* (FCO) of **FocalPolicy**. Unlike the standard setting where both time- and frequency-domain losses are defined on the same horizon, FCO uses a frequency-domain loss on a macro-trajectory of length $L = NH$ and a time-domain loss on a prefix sub-trajectory of length $H$ (steps 1:$H$). We establish two key results: (i) **Non-conflicting gradients**: the spectral and prefix time losses produce aligned (non-opposing) gradient signals, with strictly positive cosine similarity at the prediction level (and positive cosine similarity in parameter space under mild coupling conditions); (ii) **Spectral advantage beyond Parseval**: although Parseval's theorem guarantees energy equivalence between time and frequency representations on the same length-$L$ signal, we show that the spectral parameterization makes low-frequency deviations more salient via gradient concentration, improving the numerical detectability of global motion trends without requiring explicit low-frequency re-weighting.

### A.1. Definitions and Preliminaries

For clarity, we consider a single action dimension and denote the expert and predicted macro-trajectories by $\mathbf{m} \in \mathbb{R}^L$ and $\hat{\mathbf{m}}(\theta) \in \mathbb{R}^L$, respectively, where $L = NH$. Define the macro-trajectory error and Jacobian:

$$\mathbf{e} := \hat{\mathbf{m}}(\theta) - \mathbf{m}, \qquad \mathbf{J} := \frac{\partial \hat{\mathbf{m}}}{\partial \theta} \in \mathbb{R}^{L \times |\theta|}. \tag{16}$$

Let $\mathbf{D} \in \mathbb{R}^{L \times L}$ denote the length-$L$ DCT-II matrix with orthonormal normalization (Eq. 5), so that $\mathbf{D}^\top \mathbf{D} = \mathbf{I}_L$. The (length-$L$) full-band frequency-domain loss is

$$\mathcal{L}_{\text{freq}}^{(L)}(\theta) := \|\mathbf{D}\hat{\mathbf{m}}(\theta) - \mathbf{D}\mathbf{m}\|_2^2 = \|\mathbf{D}\mathbf{e}\|_2^2. \tag{17}$$

The time-domain loss is defined on the first $H$ steps of the macro-trajectory (steps 1:$H$). Let $\mathbf{S} \in \mathbb{R}^{H \times L}$ be a fixed selection matrix that extracts this supervised prefix, for example $\mathbf{S} = [\mathbf{I}_H \ \mathbf{0}_{H \times (L-H)}]$. Define

$$\mathbf{e}_H := \mathbf{S}\mathbf{e} \in \mathbb{R}^H, \qquad \mathcal{L}_{\text{time}}^{(H)}(\theta) := \|\mathbf{e}_H\|_2^2 = \|\mathbf{S}\mathbf{e}\|_2^2. \tag{18}$$

We assume $\mathbf{S}$ is a row-submatrix of $\mathbf{I}_L$ (i.e., it selects $H$ time indices without reweighting), hence $\mathbf{P} := \mathbf{S}^\top \mathbf{S} \in \mathbb{R}^{L \times L}$ is an orthogonal projector with $\mathbf{P}^\top = \mathbf{P}$ and $\mathbf{P}^2 = \mathbf{P}$.

**Theorem A.1** (Parseval's Theorem for Orthonormal DCT). *For any* $\mathbf{x} \in \mathbb{R}^L$, $\|\mathbf{x}\|_2^2 = \|\mathbf{D}\mathbf{x}\|_2^2$. *Equivalently,* $\mathbf{D}^\top \mathbf{D} = \mathbf{I}_L$.

### A.2. Gradient Compatibility

We first establish a *strictly positive* cosine similarity at the *prediction level* (w.r.t. $\hat{\mathbf{m}}$), and then provide a mild sufficient condition for positive cosine similarity in the *parameter space* (w.r.t. $\theta$).

**Gradients w.r.t. predictions $\hat{\mathbf{m}}$.** By Theorem A.1, $\mathcal{L}_{\text{freq}}^{(L)}(\theta) = \|\mathbf{e}\|_2^2$. Thus,

$$\nabla_{\hat{\mathbf{m}}} \mathcal{L}_{\text{freq}}^{(L)} = 2\mathbf{e}, \qquad \nabla_{\hat{\mathbf{m}}} \mathcal{L}_{\text{time}}^{(H)} = 2\mathbf{P}\mathbf{e}. \tag{19}$$

**Proposition A.2** (Strictly Positive Cosine Similarity in Prediction Space). *Whenever the short-window error is non-zero (i.e., $\mathbf{Pe} \neq \mathbf{0}$), we have*

$$\cos\left(\nabla_{\hat{\mathbf{m}}} \mathcal{L}_{freq}^{(L)}, \nabla_{\hat{\mathbf{m}}} \mathcal{L}_{time}^{(H)}\right) = \frac{\langle \mathbf{e}, \mathbf{Pe} \rangle}{\|\mathbf{e}\|_2 \|\mathbf{Pe}\|_2} = \frac{\|\mathbf{Pe}\|_2}{\|\mathbf{e}\|_2} > 0. \tag{20}$$

*Therefore, the macro-scale spectral term never induces an opposing supervision direction for the prefix time objective at the prediction level.*

*Proof.* Since $\mathbf{P}$ is an orthogonal projector, $\langle \mathbf{e}, \mathbf{Pe} \rangle = \mathbf{e}^\top \mathbf{Pe} = (\mathbf{Pe})^\top (\mathbf{Pe}) = \|\mathbf{Pe}\|_2^2$. Substituting yields Eq. (20). $\square$

**Gradients w.r.t. parameters $\theta$.** Applying the chain rule,

$$\nabla_\theta \mathcal{L}_{\text{freq}}^{(L)} = 2\mathbf{J}^\top \mathbf{e}, \qquad \nabla_\theta \mathcal{L}_{\text{time}}^{(H)} = 2\mathbf{J}^\top \mathbf{Pe}. \tag{21}$$

Partition $\mathbf{e} = [\mathbf{e}_{\mathcal{S}}; \mathbf{e}_{\mathcal{R}}]$ and $\mathbf{J} = [\mathbf{J}_{\mathcal{S}}; \mathbf{J}_{\mathcal{R}}]$, where $\mathcal{S}$ corresponds to the prefix indices $1{:}H$. Then

$$\nabla_\theta \mathcal{L}_{\text{freq}}^{(L)} = 2\left(\mathbf{J}_{\mathcal{S}}^\top \mathbf{e}_{\mathcal{S}} + \mathbf{J}_{\mathcal{R}}^\top \mathbf{e}_{\mathcal{R}}\right), \qquad \nabla_\theta \mathcal{L}_{\text{time}}^{(H)} = 2\mathbf{J}_{\mathcal{S}}^\top \mathbf{e}_{\mathcal{S}}. \tag{22}$$

**Assumption A.3** (Non-adversarial temporal coupling). The gradient contribution from the remaining (unobserved in time loss) steps is not adversarial to the supervised short-window contribution:

$$\langle \mathbf{J}_{\mathcal{R}}^\top \mathbf{e}_{\mathcal{R}}, \; \mathbf{J}_{\mathcal{S}}^\top \mathbf{e}_{\mathcal{S}} \rangle \geq 0. \tag{23}$$

**Proposition A.4** (Positive Cosine Similarity in Parameter Space). *Under Assumption A.3, whenever $\nabla_\theta \mathcal{L}_{time}^{(H)} \neq \mathbf{0}$,*

$$\left\langle \nabla_\theta \mathcal{L}_{freq}^{(L)}, \nabla_\theta \mathcal{L}_{time}^{(H)} \right\rangle \geq \left\| \nabla_\theta \mathcal{L}_{time}^{(H)} \right\|_2^2 > 0, \tag{24}$$

*and hence* $\cos\left(\nabla_\theta \mathcal{L}_{freq}^{(L)}, \nabla_\theta \mathcal{L}_{time}^{(H)}\right) > 0$.

*Proof.* By decomposition,

$$\begin{aligned}
\left\langle \nabla_\theta \mathcal{L}_{\text{freq}}^{(L)}, \nabla_\theta \mathcal{L}_{\text{time}}^{(H)} \right\rangle &= 4 \left\langle \mathbf{J}_{\mathcal{S}}^\top \mathbf{e}_{\mathcal{S}} + \mathbf{J}_{\mathcal{R}}^\top \mathbf{e}_{\mathcal{R}}, \; \mathbf{J}_{\mathcal{S}}^\top \mathbf{e}_{\mathcal{S}} \right\rangle \\
&= 4 \left\| \mathbf{J}_{\mathcal{S}}^\top \mathbf{e}_{\mathcal{S}} \right\|_2^2 + 4 \left\langle \mathbf{J}_{\mathcal{R}}^\top \mathbf{e}_{\mathcal{R}}, \; \mathbf{J}_{\mathcal{S}}^\top \mathbf{e}_{\mathcal{S}} \right\rangle. \tag{25}
\end{aligned}$$

Assumption A.3 makes the second term non-negative, yielding the claim. $\square$

*Remark* A.5 (A sufficient structural condition). A sufficient structural condition for Assumption A.3 is $\mathbf{J}_{\mathcal{R}} \mathbf{J}_{\mathcal{S}}^\top = \mathbf{0}$, i.e., the parameter sensitivities of the supervised prefix window and the remaining steps are orthogonal (e.g., temporally separated heads). Even when not exact, this orthogonality is often approximately satisfied, leading to empirically positive cosine similarity.

*Remark* A.6 (Why Parseval does not contradict empirical gains). Parseval's theorem implies that the *macro-scale* unweighted full-band spectral loss $\|\mathbf{De}\|_2^2$ equals the *macro-scale* time-domain loss $\|\mathbf{e}\|_2^2$ *on the same length-$L$ signal*. In FCO, however, the time-domain term is defined only on a *prefix window* of length $H$, whereas the spectral term supervises the full macro-trajectory of length $L = NH$. Therefore, the two objectives are not equivalent, and the spectral term provides additional full-trajectory supervisory signal that can improve optimization and long-range consistency.

### A.3. Spectral Sensitivity Analysis

A core motivation for using frequency-domain supervision is its ability to capture overall motion trend. While Parseval's theorem guarantees energy equivalence on the *same* length-$L$ signal (i.e., $\|\mathbf{e}\|_2 = \|\mathbf{De}\|_2$ for orthonormal $\mathbf{D}$), it does not imply equivalence in *optimization dynamics* in our setting: FCO pairs a full-trajectory spectral loss with a prefix-window time loss, and, moreover, the DCT representation can *concentrate* low-frequency (trend-level) errors into a small set of coefficients, making such errors more numerically salient under noisy stochastic optimization.

Let $\mathbf{E} = \mathbf{De}$ denote the DCT coefficients of the error. The unweighted full-band spectral loss can be written as

$$\mathcal{L}_{\text{freq}}^{(L)} = \frac{1}{2}\|\mathbf{E}\|_2^2 = \frac{1}{2}\sum_{k=0}^{L-1} E_k^2, \qquad \nabla_{\mathbf{E}}\mathcal{L}_{\text{freq}}^{(L)} = \mathbf{E}. \tag{26}$$

This mode-wise decomposition makes explicit that each frequency component contributes independently, and that errors dominated by a few low-frequency modes yield *sparse, concentrated* gradients in coefficient space.

**Proposition A.7** (Gradient Concentration and Sensitivity Gain for Macroscopic Drift)**.** *Consider a macroscopic drift error modeled as a constant DC offset $e_n = c$ for all $n \in \{1,\dots,L\}$, where $|c| \ll 1$. Let $\phi_0 = \frac{1}{\sqrt{L}}\mathbf{1}$ denote the DC DCT basis vector. Then the error concentrates entirely on the DC coefficient:*

$$E_0 = \langle \mathbf{e}, \phi_0 \rangle = \sqrt{L}\,c, \qquad E_k = 0 \ \text{for} \ k > 0, \tag{27}$$

*and hence the coefficient-space gradient is sparse:*

$$(\nabla_{\mathbf{E}}\mathcal{L}_{freq}^{(L)})_k = \begin{cases} \sqrt{L}\,c, & k = 0, \\ 0, & k > 0. \end{cases} \tag{28}$$

*Consequently, the peak gradient magnitude in coefficient space satisfies*

$$\|\nabla_{\mathbf{E}}\mathcal{L}_{freq}^{(L)}\|_\infty = \sqrt{L}|c|. \tag{29}$$

*In contrast, in the time domain, the per-timestep gradient for the full-trajectory quadratic loss $\mathcal{L}_{time}^{(L)} = \frac{1}{2}\|\mathbf{e}\|_2^2$ is uniformly distributed:*

$$\nabla_{\hat{m}_n}\mathcal{L}_{time}^{(L)} = e_n = c, \quad \forall n, \qquad \|\nabla_{\hat{\mathbf{m}}}\mathcal{L}_{time}^{(L)}\|_\infty = |c|. \tag{30}$$

*Therefore, for coherent macroscopic drift, the DCT coefficient space exhibits a sensitivity gain of $\sqrt{L}$ in the peak gradient magnitude:*

$$\text{Gain} = \frac{\|\nabla_{\mathbf{E}}\mathcal{L}_{freq}^{(L)}\|_\infty}{\|\nabla_{\hat{\mathbf{m}}}\mathcal{L}_{time}^{(L)}\|_\infty} = \sqrt{L}. \tag{31}$$

**Implication for numerical optimization (SNR viewpoint).** In stochastic gradient optimization, gradient estimates are often corrupted by noise $\eta$ (e.g., mini-batching noise or interference from other objective terms). When the macroscopic drift $|c|$ is small, the uniform time-domain gradient magnitude $|c|$ can be dominated by noise, leading to ineffective updates. In the DCT coefficient space, however, the same drift produces a concentrated signal of magnitude $\sqrt{L}|c|$ on a small set of low-frequency coefficients (here, only the DC term), improving the effective signal-to-noise ratio. In particular, even when $|c| \lesssim \|\eta\|_\infty$, the condition $\sqrt{L}|c| \gtrsim \|\eta\|_\infty$ can still hold for moderate $L$, enabling the optimizer to reliably detect and correct global trend errors. This provides a numerical explanation for why unweighted full-band spectral supervision can better capture macroscopic motion structure without requiring explicit low-frequency re-weighting.

**Why explicit low-frequency weighting is not superior.** For completeness, consider a weighted spectral objective with $\mathbf{W} = \mathrm{diag}(w_0, \dots, w_{L-1})$:

$$\mathcal{L}_{\text{w-spec}} = \frac{1}{2}\|\mathbf{WE}\|_2^2 = \frac{1}{2}\sum_{k=0}^{L-1} w_k^2 E_k^2. \tag{32}$$

Its gradients satisfy

$$\nabla_{\mathbf{E}}\mathcal{L}_{\text{w-spec}} = \mathbf{W}^2\mathbf{E}, \qquad \nabla_{\hat{\mathbf{m}}}\mathcal{L}_{\text{w-spec}} = \mathbf{D}^\top \mathbf{W}^2 \mathbf{D}\,\mathbf{e}. \tag{33}$$

If $w_k$ decays with $k$ (emphasizing low frequencies), then $\mathbf{D}^\top \mathbf{W}^2 \mathbf{D}$ shapes the time-domain gradient toward a low-pass update: it strengthens corrections to smooth global structure but suppresses higher-frequency corrections. While this can help when errors are purely trend-dominated, diverse manipulation behaviors often require fine-scale, high-frequency adjustments (e.g., contact transitions and rapid micro-corrections). Over-emphasizing low frequencies can therefore under-train these components, explaining why explicit band weighting is not better than the unweighted full-band spectral loss in practice.

## B. Theoretical Analysis of Propagation Efficiency

In this appendix, we analyze the proposed *Locally Anchored Sampling* (LAS) strategy in **FocalPolicy**. We formalize a metric of *Target-signal Propagation Efficiency* that measures how faithfully a consistency-based update at a student time $\tau$ matches an ideal supervised update, and prove that biasing the teacher (anchored) time $r$ towards the terminal region improves this efficiency under a standard monotonicity assumption. Consequently, LAS promotes more accurate target-signal propagation during consistency training.

### B.1. Preliminaries and Definitions

We use the same notation as Sec. 4.2. Let $\tau$ denote the student flow time and $r$ denote the teacher (anchored) time. Along the OT path (Eq. (11)), the interpolated macro-trajectories are

$$\mathbf{M}_\tau = (1 - \tau)\mathbf{M}_0 + \tau\mathbf{M}_1, \qquad \mathbf{M}_r = (1 - r)\mathbf{M}_0 + r\mathbf{M}_1, \tag{34}$$

where $\mathbf{M}_0$ is Gaussian noise and $\mathbf{M}_1$ is the expert macro-trajectory. Conditioned on the observation $o_t$, the model predicts velocities $v_\theta(\mathbf{M}_\tau, \tau, o_t)$ and the EMA target network predicts $v_{\theta^-}(\mathbf{M}_r, r, o_t)$. In consistency-based flow matching, for a given student timestep $\tau$, we sample a teacher timestep $r$ biased towards the terminal region (typically $r \geq \tau$). The standard consistency loss (simplified for a single sample) is given by:

$$\mathcal{L}_{\text{cons}}(\theta; \tau, r) = \|v_\theta(\mathbf{M}_\tau, \tau, o_t) - \text{sg}[v_{\theta^-}(\mathbf{M}_r, r, o_t)]\|_2^2, \tag{35}$$

where $\text{sg}[\cdot]$ denotes stop-gradient. The resulting (velocity-space) consistency gradient at time $\tau$ is

$$g_{\text{cons}}(\tau, r) := \nabla_{v_\theta(\mathbf{M}_\tau, \tau, o_t)} \mathcal{L}_{\text{cons}} = 2\Big(v_\theta(\mathbf{M}_\tau, \tau, o_t) - v_{\theta^-}(\mathbf{M}_r, r, o_t)\Big). \tag{36}$$

Under the OT construction used in flow matching, the optimal target velocity is constant:

$$u^*(\mathbf{M}_\tau, \tau, o_t) = \mathbf{M}_1 - \mathbf{M}_0. \tag{37}$$

Thus the ideal supervised gradient at time $\tau$ is

$$g_{\text{sup}}(\tau) := \nabla_{v_\theta(\mathbf{M}_\tau, \tau, o_t)} \|v_\theta(\mathbf{M}_\tau, \tau, o_t) - u^*(\mathbf{M}_\tau, \tau, o_t)\|^2 = 2\Big(v_\theta(\mathbf{M}_\tau, \tau, o_t) - (\mathbf{M}_1 - \mathbf{M}_0)\Big). \tag{38}$$

**Definition B.1** (Target Signal Propagation Efficiency). *For a fixed student time $\tau$, let $p(r|\tau)$ denote the sampling distribution of the teacher time $r$. The propagation efficiency $\mathcal{E}(\tau)$ is defined as the negative expected mean squared error between the consistency gradient and the ideal supervised gradient:*

$$\mathcal{E}(\tau) := -\mathbb{E}_{r \sim p(r|\tau)}\left[\|g_{\text{cons}}(\tau, r) - g_{\text{sup}}(\tau)\|_2^2\right]. \tag{39}$$

Substituting the gradient definitions, we can simplify the efficiency term:

$$\|g_{\text{cons}}(\tau, r) - g_{\text{sup}}(\tau)\|_2^2 = \|2(v_\theta(\tau) - v_{\theta^-}(r)) - 2(v_\theta(\tau) - u^*)\|_2^2$$
$$= 4\|u^* - v_{\theta^-}(\mathbf{M}_r, r, o_t)\|_2^2, \tag{40}$$

where we abbreviate $v_\theta(\tau) := v_\theta(\mathbf{M}_\tau, \tau, o_t)$, $v_{\theta^-}(r) := v_{\theta^-}(\mathbf{M}_r, r, o_t)$, and $u^* := \mathbf{M}_1 - \mathbf{M}_0$. Remarkably, the student model's prediction $v_\theta(\tau)$ cancels out, indicating that the propagation efficiency depends solely on the quality of the teacher's target estimation at time $r$. By construction of the OT path (Eq. (34)), the optimal velocity field implies a straight trajectory, thus $u^*(\mathbf{M}_\tau, \tau) = \mathbf{M}_1 - \mathbf{M}_0$ is constant for all $\tau \in [0, 1]$. Thus, the metric simplifies to the teacher's prediction error:

$$\mathcal{E}(\tau) \propto -\mathbb{E}_{r \sim p(r|\tau)}\left[\|v_{\theta^-}(\mathbf{M}_r, r, o_t) - (\mathbf{M}_1 - \mathbf{M}_0)\|_2^2\right]. \tag{41}$$

Eq. (41) shows that maximizing $\mathcal{E}(\tau)$ is equivalent to minimizing the expected teacher prediction error evaluated at the sampled teacher time $r$.

## B.2. Proof of Superiority

We now compare the proposed Locally Anchored Sampling against a standard baseline (e.g., uniform sampling or local neighborhood sampling).

**Assumption B.2** (Monotone Teacher Error Towards the Terminal). Let

$$\epsilon(s) := \mathbb{E}\Big[\|v_{\theta-}(\mathbf{M}_s, s, o_t) - (\mathbf{M}_1 - \mathbf{M}_0)\|_2^2\Big]. \tag{42}$$

We assume $\epsilon(s)$ is strictly decreasing as $s$ approaches the terminal boundary:

$$\forall s_1, s_2 \in [0, 1], \quad s_1 < s_2 \Rightarrow \epsilon(s_1) > \epsilon(s_2). \tag{43}$$

**Definition B.3** (First-Order Stochastic Dominance). For a fixed $\tau$, we say $p_{\text{LAS}}(r|\tau)$ first-order stochastically dominates $p_{\text{base}}(r|\tau)$ on $[\tau, 1]$, denoted $p_{\text{LAS}} \succeq_{\text{FSD}} p_{\text{base}}$, if for all $a \in [\tau, 1]$,

$$\Pr_{r \sim p_{\text{LAS}}} (r \geq a \,|\, \tau) \geq \Pr_{r \sim p_{\text{base}}} (r \geq a \,|\, \tau), \tag{44}$$

with strict inequality for some $a$.

**Proposition B.4.** *Under Assumption B.2, if $p_{LAS}(r|\tau) \succeq_{\text{FSD}} p_{base}(r|\tau)$ for all $\tau$, the Locally Anchored Sampling strategy yields strictly higher propagation efficiency than any sampling strategy that places probability mass further from the terminal state.*

*Proof.* From Eq. (41), for any sampling strategy $\pi$ we can write the propagation efficiency (up to a positive constant factor) as

$$\mathcal{E}_\pi(\tau) \propto -\mathbb{E}_{r \sim p_\pi(r|\tau)}\big[\epsilon(r)\big], \tag{45}$$

where $\epsilon(r)$ is defined in Assumption B.2 and is strictly decreasing on $[0, 1]$.

We now compare $\pi \in \{\text{LAS}, \text{base}\}$ for a fixed $\tau$. By Definition B.3, the condition $p_{\text{LAS}}(\cdot|\tau) \succeq_{\text{FSD}} p_{\text{base}}(\cdot|\tau)$ means that $p_{\text{LAS}}$ assigns (weakly) more probability mass to larger teacher times:

$$\Pr_{r \sim p_{\text{LAS}}} (r \geq a \,|\, \tau) \geq \Pr_{r \sim p_{\text{base}}} (r \geq a \,|\, \tau), \qquad \forall a \in [\tau, 1], \tag{46}$$

with strict inequality for at least one threshold $a$. Equivalently, letting $F_{\text{LAS}}(a|\tau)$ and $F_{\text{base}}(a|\tau)$ denote the conditional CDFs of $r$, this is the same as

$$F_{\text{LAS}}(a|\tau) \leq F_{\text{base}}(a|\tau), \qquad \forall a \in [\tau, 1], \tag{47}$$

with strict inequality on a set of nonzero measure.

Since $\epsilon(\cdot)$ is strictly decreasing (Assumption B.2), the random variable $\epsilon(r)$ is therefore stochastically *smaller* when $r$ is stochastically larger. Formally, a standard consequence of first-order stochastic dominance states that for any strictly decreasing measurable function $\phi$,

$$r_{\text{LAS}} \succeq_{\text{FSD}} r_{\text{base}} \implies \mathbb{E}\big[\phi(r_{\text{LAS}})\big] < \mathbb{E}\big[\phi(r_{\text{base}})\big], \tag{48}$$

whenever the dominance is strict. Applying this result with $\phi(\cdot) = \epsilon(\cdot)$ yields

$$\mathbb{E}_{r \sim p_{\text{LAS}}(r|\tau)}[\epsilon(r)] < \mathbb{E}_{r \sim p_{\text{base}}(r|\tau)}[\epsilon(r)]. \tag{49}$$

Finally, substituting Eq. (49) into Eq. (45) and noting the leading negative sign, we obtain

$$\mathcal{E}_{\text{LAS}}(\tau) > \mathcal{E}_{\text{base}}(\tau), \tag{50}$$

In summary, by shifting the sampling distribution towards the terminal region where the teacher is more reliable, Locally Anchored Sampling effectively reduces the variance of the supervision signal, leading to higher propagation efficiency. $\square$

*Table 6.* **Ablation of anchor time $r$ configurations** in LAS. We compare the sampling distributions of $r$ with different $\mu_r$ and $\sigma_r$ settings. The configuration ($\mu_r = 4.0, \sigma_r = 1.6$, biased toward 1.0) represents our optimal setting, which yields the best performance across most tasks.

| Config. of $r$ | Adroit | | | Meta-World | | | | Average |
|---|---|---|---|---|---|---|---|---|
| | Hammer | Door | Pen | Sweep-Into | HAND-INSERT | PICK-PLACE | DISASSEMBLE | |
| $\mu_r = 0.4, \sigma_r = 0.5 \; (\approx 0.6)$ | $14.0 \pm 4.4$ | $17.7 \pm 6.7$ | $35.0 \pm 7.5$ | $12.3 \pm 3.1$ | $8.3 \pm 1.5$ | $1.0 \pm 1.0$ | $9.0 \pm 1.0$ | 13.9 |
| $\mu_r = 1.4, \sigma_r = 0.5 \; (\approx 0.8)$ | $80.7 \pm 3.5$ | $59.7 \pm 5.9$ | $63.0 \pm 3.6$ | $32.0 \pm 22.3$ | $16.7 \pm 2.5$ | $29.0 \pm 3.6$ | $52.3 \pm 9.3$ | 47.6 |
| **Ours** ($\mu_r = 4.0, \sigma_r = 1.6$) | $\mathbf{100.0 \pm 0.0}$ | $\mathbf{63.2 \pm 5.3}$ | $\mathbf{67.5 \pm 1.3}$ | $\mathbf{70.7 \pm 22.0}$ | $\mathbf{29.3 \pm 16.2}$ | $\mathbf{70.7 \pm 4.2}$ | $\mathbf{86.7 \pm 1.5}$ | **69.7** |

# C. Experiment Details

## C.1. Hyperparameters Setup

This section details the hyper-parameter configurations used for training and evaluating the proposed FocalPolicy. Table. 7 summarizes the complete set of hyper-parameters used in our experiments. For all baseline methods, including DP3, FlowPolicy, and FreqPolicy, we use the default hyper-parameter settings to avoid introducing unintended advantages. For both the Adroit and Meta-World benchmarks, all models are trained for 3000 training iterations to ensure convergence and a fair comparison of learning dynamics.

**Hyperparameters for FocalPolicy in LAS** ($\mu = 4.0, \sigma = 1.6$). Our theoretical analysis (Proposition B.4) indicates that the sampling distribution should be strictly skewed towards the terminal time $r = 1$ to maximize propagation efficiency while maintaining non-zero variance for robustness. We employ a Logit-Normal distribution $r = \sigma(x), x \sim \mathcal{N}(\mu, \sigma^2)$.

- Choice of $\mu = 4.0$: This maps to a median time of $r \approx 0.982$, ensuring that the majority of consistency supervision comes from the near-terminal region where the teacher's prediction error $\epsilon(r)$ is minimal.

- Choice of $\sigma = 1.6$: This standard deviation provides a heavy left tail in the probability density. Specifically, it ensures that approximately $95\%$ of the sampling mass is concentrated in the high-fidelity region $r \in [0.5, 1.0]$, while retaining a small probability for $r < 0.5$ to enforce global consistency constraints preventing overfitting to the terminal state.

To further investigate the impact of different anchoring strategies, we compare three representative parameter configurations for the time $r$: $\mu_r = 1.4, \sigma_r = 0.5$ (anchored near 0.8), $\mu_r = 0.4, \sigma_r = 0.5$ (anchored near 0.6), and our optimal setting $\mu_r = 4.0, \sigma_r = 1.6$ (biased toward 1.0). The resulting sampling distributions for the flow timesteps $\tau$ and $r$, derived from 50,000 random samples, are visualized in Figure 8.

We compared the performance of our model under three different sampling parameter configurations across three Adroit tasks and four MetaWorld tasks, as summarized in Table 6. The results demonstrate a significant performance degradation as the anchor time $r$ shifts away from the terminal region; specifically, the average success rate plummets from 69.7% to 13.9% when the anchor center moves from approximately 1.0 to 0.6. These observations directly validate Assumption B.2, confirming that the teacher model's predictive accuracy diminishes significantly as the anchor time $r$ moves toward earlier, more stochastic flow stages. As elucidated in our theoretical analysis in Appendix B, this degradation in teacher reliability leads to a rapid decay of the target-signal propagation efficiency $\mathcal{E}(\tau)$, because the consistency gradient $g_{\mathrm{cons}}$ fails to faithfully approximate the ideal supervised gradient $g_{\mathrm{sup}}$. Such empirical findings provide strong evidence that local anchoring within the high-fidelity terminal region is an indispensable condition for providing stable, low-variance supervision. This mechanism plays a decisive role in enabling FocalPolicy to effectively capture the complex distributions inherent in the longer macro-trajectories.

## C.2. Implementation details

**Network Architecture and Hardware.** Following prior work (Chi et al., 2023; Ze et al., 2024; Jia et al., 2024; Lu et al., 2024; Zhang et al., 2025a; Su et al., 2025a), we adopt a standard 1D CNN-based U-Net architecture as the backbone of FocalPolicy to enable fair and well-controlled comparisons with existing visuomotor policies. FocalPolicy is primarily designed for 3D visuomotor manipulation. We encode the input 3D point clouds with a lightweight MLP to obtain compact feature representations, which are then fused with the action-generation backbone. This design provides a favorable trade-off between representational capacity and computational efficiency, and is consistent with the point-cloud encoders commonly

*Table 7.* **Hyperparameters setup for FOCAL.**

| Hyperparameter | Value |
|---|---|
| Horizon ($L$) | 12 |
| Action steps ($H$) | 4 |
| Observation steps ($n_{obs}$) | 2 |
| Pointcloud feature dim | 64 |
| State mlp size | 64 |
| Frequency weight ($\lambda_{freq}$) | 1.0e-4 |
| Anchor $r$ mean ($\mu_r$) | 4.0 |
| Anchor $r$ std ($\sigma_r$) | 1.6 |
| Condition type | FiLM |
| U-Net down dims | [512, 1024, 2048] |
| Kernel size | 5 |
| Number of groups ($n_{groups}$) | 8 |
| Pointnet type | MLP |
| Batch size | 128 |
| Number of epochs | 3000 |
| Optimizer | AdamW |
| Betas ($\beta_1, \beta_2$) | [0.95, 0.999] |
| Learning rate | 1.0e-4 |
| Weight decay | 1.0e-6 |
| LR scheduler | Cosine (500 steps warmup) |
| EMA max value | 0.9999 |

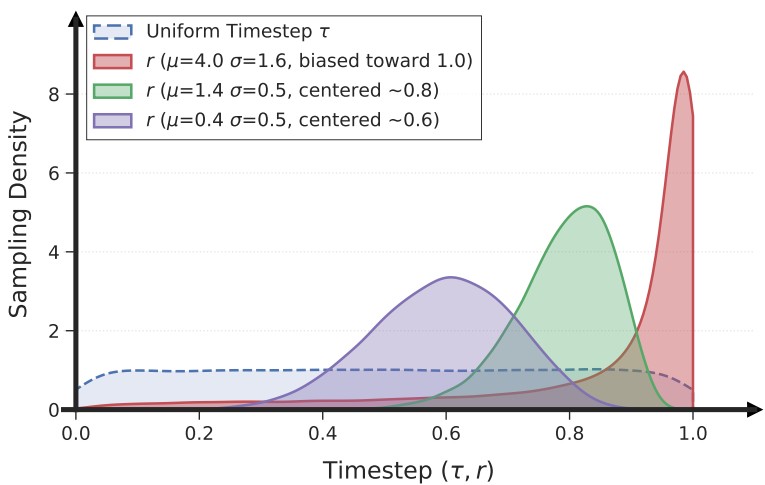

*Figure 8.* Visualization of the sampling distributions in LAS. We compare standard uniform sampling (dashed blue curve) against our **Locally Anchored Time Sampling** (solid red curve) for flow timepoints. By biasing the anchor time $r$ towards the terminal boundary ($\tau \to 1$), FOCAL significantly enhances the target signal propagation efficiency compared to uniform sampling, thereby stabilizing training for complex multi-chunk action distributions.

used in prior 3D manipulation policies (Ze et al., 2024; Zhang et al., 2025a; Su et al., 2025a). All experiments are conducted on NVIDIA RTX 4090, RTX 3090, and A6000 GPUs. The complete training process is summarized in **Algorithm 1**.

**Training Time and Computational Costs.** We benchmark the computational efficiency of FocalPolicy against DP3 and FlowPolicy using a workstation equipped with an Intel Core i9-14900KF CPU and an NVIDIA GeForce RTX 4090 GPU. The evaluation is conducted across four MetaWorld tasks spanning a spectrum of difficulty levels: Reach Wall (Easy), Sweep Into (Medium), Pick Place (Hard), and Disassemble (Very Hard). Detailed metrics are summarized in Table 8. The results indicate that FocalPolicy achieves the lowest evaluation latency (26.22s). This efficiency stems from our hierarchical

**Algorithm 1** FocalPolicy Training Algorithm

**Require**: Dataset $\mathcal{D}$, Train Horizon $L$, Execution Horizon $H$, Anchor params $\mu_r, \sigma_r$, Weight $\lambda$.

**Initialize**: Policy network $\theta$, Target network $\theta^- \leftarrow \theta$.

▷ **Sampling Step**:

$(o_t, \mathbf{M}_t) \sim \mathcal{D}; \mathbf{M}_0 \sim \mathcal{N}(\mathbf{0}, \mathbf{I}); \epsilon \sim \mathcal{N}(0,1); \tau \sim \mathcal{U}[0,1]; r = \text{sigmoid}(\mu_r + \sigma_r \cdot \epsilon);$

▷ **Construct OT interpolants**

$\mathbf{M}_\tau = (1 - \tau)\mathbf{M}_0 + \tau\mathbf{M}_t; \mathbf{M}_r = (1 - r)\mathbf{M}_0 + r\mathbf{M}_t;$

▷ **Flow Prediction**:

$v_\theta^\tau \leftarrow v_\theta(\mathbf{M}_\tau, \tau, o_t); v_{\theta^-}^r \leftarrow v_{\theta^-}(\mathbf{M}_r, r, o_t);$

$\hat{\mathbf{M}}^\tau \leftarrow f_\theta(\mathbf{M}_\tau, \tau, o_t) = \mathbf{M}_\tau + (1 - \tau)\, v_\theta^\tau;$

$\hat{\mathbf{M}}^r \leftarrow f_{\theta^-}(\mathbf{M}_r, r, o_t) = \mathbf{M}_r + (1 - r)\, v_{\theta^-}^r;$

▷ **Dual-Horizon Loss**:

$\mathcal{L}_{\text{time}} = \|(\hat{\mathbf{M}}^\tau)_{1:H} - (\hat{\mathbf{M}}^r)_{1:H}\|^2; \mathcal{L}_{\text{freq}} = \|\text{DCT}(\hat{\mathbf{M}}^\tau) - \text{DCT}(\mathbf{M}_t)\|^2;$

$\mathcal{L}_{\text{FCO}} = \mathcal{L}_{\text{time}} + \lambda\mathcal{L}_{\text{freq}};$

▷ **Optimization**:

$\theta \leftarrow \theta - \eta\nabla_\theta\mathcal{L}_{\text{FCO}}; \theta^- \leftarrow \beta\theta^- + (1 - \beta)\theta;$

**Return**: Optimized Policy Parameters $\theta$.

*Table 8.* **Computational Costs and Performance Comparison.** Metrics are measured on a single workstation equipped with an **Intel Core i9-14900KF CPU** and an **NVIDIA GeForce RTX 4090 GPU**.

| Method | GPU Memory | Train Time | Eval Time (s) $\downarrow$ | Success Rate (%) $\uparrow$ |
|---|---|---|---|---|
| DP3 | 7769 MB | 1.3 h | 42.49 | 61.3 |
| FlowPolicy | 7377 MB | 1.4 h | 28.92 | 59.6 |
| **FocalPolicy** | 8035 MB | 1.5 h | 26.22 | 78.6 |

design: compared to FlowPolicy (28.92s), FocalPolicy utilizes a longer prediction horizon per inference; compared to DP3 (42.49s), it requires significantly fewer Number of Function Evaluations (NFE). Although the training time and GPU memory footprint of FocalPolicy (8035MB) are slightly higher than those of FlowPolicy (7377MB) and DP3 (7769MB) due to additional frequency-domain transformations and the modeling of extended macro-trajectories, this marginal increase in resource consumption is well-justified by a substantial gain of approximately 20% in average success rate.

**Module Generalizability setups.** To evaluate the plug-and-play potential of our modules, we integrate *Foresight Composite Objective* (FCO) and *Locally Anchored Sampling* (LAS) into representative baselines. First, since **DP3** relies on diffusion sampling steps, it is not directly compatible with our flow-based time sampling (**LAS**). We therefore only integrate **FCO** into DP3, denoted as **DP3 w. Ours**. Concretely, we apply the frequency-domain loss to DP3's 16-step predicted trajectory, while applying the time-domain loss on the 8-step execution chunk. Second, for **FlowPolicy**, we integrate our **LAS** module, denoted as **FlowPolicy w. Ours**. In this variant, we replace FlowPolicy's original time sampling with our LAS. Since our sampling strategy differs from standard consistency flow matching, we remove FlowPolicy's original segmented time-domain loss to align with the LAS formulation. Note that we do not integrate **FCO** into FlowPolicy in this plug-and-play test. This is because FlowPolicy, in its default configuration, predicts and executes only 4 steps. This short prediction horizon lacks the multi-chunk macro-trajectory structure necessary for frequency-domain structural regularization. Integrating FCO would require altering FlowPolicy's horizon settings, which is beyond the scope of this evaluation of module compatibility.

### C.3. Simulation experiment results

To comprehensively evaluate the robustness and generalization of the FocalPolicy, we report the performance across 53 challenging manipulation tasks from the Adroit and Meta-World benchmarks. For each task, we conduct evaluations over 3 independent random seeds (0, 1, 2). We report the mean success rate (%) along with the sample standard deviation to rigorously account for stochastic variability during training and rollout. To ensure a principled and fair comparison, all evaluated methods—including our FocalPolicy and all baselines—are trained on the identical set of expert demonstrations and evaluated under a unified protocol. The exhaustive per-task results are documented in Table. 15.

*Table 9.* Supplementary evaluation on RoboMimic and LIBERO benchmarks. We report the mean success rate (0–1) across diverse and modern tasks, alongside the average inference latency (Cost).

| Method \ Task | RoboMimic | | | | LIBERO | | | Average ↑ | Cost (ms) ↓ |
|---|---|---|---|---|---|---|---|---|---|
| | Lift | Can | Square | Transport | Open-Drawer | Put-Bowl | Stack-Bowl | | |
| FlowPolicy | 0.99 | 0.98 | 0.74 | 0.44 | 0.88 | 0.55 | 0.54 | 0.73 | **5.40** |
| FreqPolicy | 0.98 | 0.90 | 0.74 | 0.38 | **1.00** | 0.70 | **0.64** | 0.76 | 18.90 |
| ACG | 0.99 | 0.99 | 0.78 | 0.50 | 0.97 | 0.46 | 0.22 | 0.70 | 9.50 |
| **Ours** | **1.00** | **1.00** | **0.82** | **0.67** | **1.00** | **0.73** | 0.45 | **0.81** | **5.40** |

**Generalization to Modern and Diverse Benchmarks.** To further validate the robustness and generalization of FocalPolicy beyond scripted heuristics, we extend our evaluation to more diverse and modern benchmarks, including **RoboMimic** (Mandlekar et al., 2022) and **LIBERO** (Liu et al., 2023). Unlike Adroit and Meta-World, RoboMimic provides expert datasets collected from human demonstrators, which introduce significant behavioral diversity and noise, posing a greater challenge for action chunking. LIBERO represents a more recent and complex suite of tasks designed to test the limits of imitation learning policies.

As summarized in Table 9, we compare FocalPolicy against several strong baselines, including **ACG** (Park et al., 2025) (integrated into FlowPolicy for a fair comparison), which is explicitly designed to address cross-chunk coherence. FocalPolicy consistently outperforms all baselines across both benchmarks. Notably, on the high-dimensional human-demonstrated RoboMimic tasks (*e.g.*, *Square* and *Transport*), FocalPolicy achieves the highest success rates, demonstrating its superior ability to model complex, multi-modal expert distributions.

Furthermore, we evaluate the computational efficiency in terms of mean inference latency. Despite its enhanced foresight and coherence, FocalPolicy maintains a low inference cost of $5.40$ ms, matching the vanilla FlowPolicy and significantly outperforming FreqPolicy ($18.90$ ms) and ACG ($9.50$ ms). This efficiency stems from our FCO formulation, which regularizes the policy during training without requiring auxiliary modules or iterative optimization during inference. These results confirm that FocalPolicy advances the state-of-the-art by providing a more effective and efficient solution for coherent long-horizon manipulation.

**Mitigation of Compounding Errors.** To evaluate the robustness of our model against compounding errors in long-trajectory prediction, we conduct a comparative analysis between FocalPolicy and the baseline FlowPolicy. Both models are trained for 3,000 epochs on the same expert training set using identical configurations: a macro-trajectory length of $L = 36$ and an execution chunk of $H = 12$. To isolate the accumulated drift inherent in temporal predictions, we perform an open-loop rollout on a held-out test set. Specifically, both policies are initialized using only the first two frames of observations from the test sequences. From this initial state, the models predict the full subsequent trajectory of 36 steps without further environmental feedback. Given that MetaWorld data is recorded in incremental coordinates, we normalize the starting position of all trajectories to the origin $(0, 0, 0)$ to facilitate a direct comparison of spatial deviation. We then visualize the predicted trajectories against the ground-truth expert demonstrations and quantify the compounding error at each timestep using the Euclidean distance of absolute positions. This setup rigorously tests the policies' ability to maintain structural coherence over long trajectories without the corrective benefit of closed-loop observations. As shown in Figure 9, to provide a more comprehensive demonstration of FocalPolicy, we compare the predicted trajectories and compounding errors against FlowPolicy on additional MetaWorld tasks, including *Pick Place*, *Pick Out of Hole*, *Sweep Into*, *Pick Place Wall Disassemble* and *Shelf Place*.

*Table 10.* Quantitative comparison of Trajectory Smoothness score ($\text{TS}_{\text{score}}$ ↓). A lower score indicates that the policy's trajectory smoothness is closer to the expert's.

| Method | Bin Picking | Disassemble | Pick Out of Hole | Pick Place | Pick Place Wall | Push Wall | Shelf Place | Sweep Into |
|---|---|---|---|---|---|---|---|---|
| FlowPolicy | 0.27 | 0.09 | 0.04 | **0.01** | 0.03 | 0.09 | 0.03 | 0.50 |
| **Ours** | **0.13** | **0.01** | **0.02** | **0.01** | **0.00** | **0.00** | **0.01** | **0.40** |

**Quantitative Evaluation of Trajectory Smoothness.** Furthermore, we calculated the Trajectory Smoothness (TS) metric introduced in (Songwei et al., 2026) to rigorously evaluate the temporal smoothness of the predicted trajectories. In this

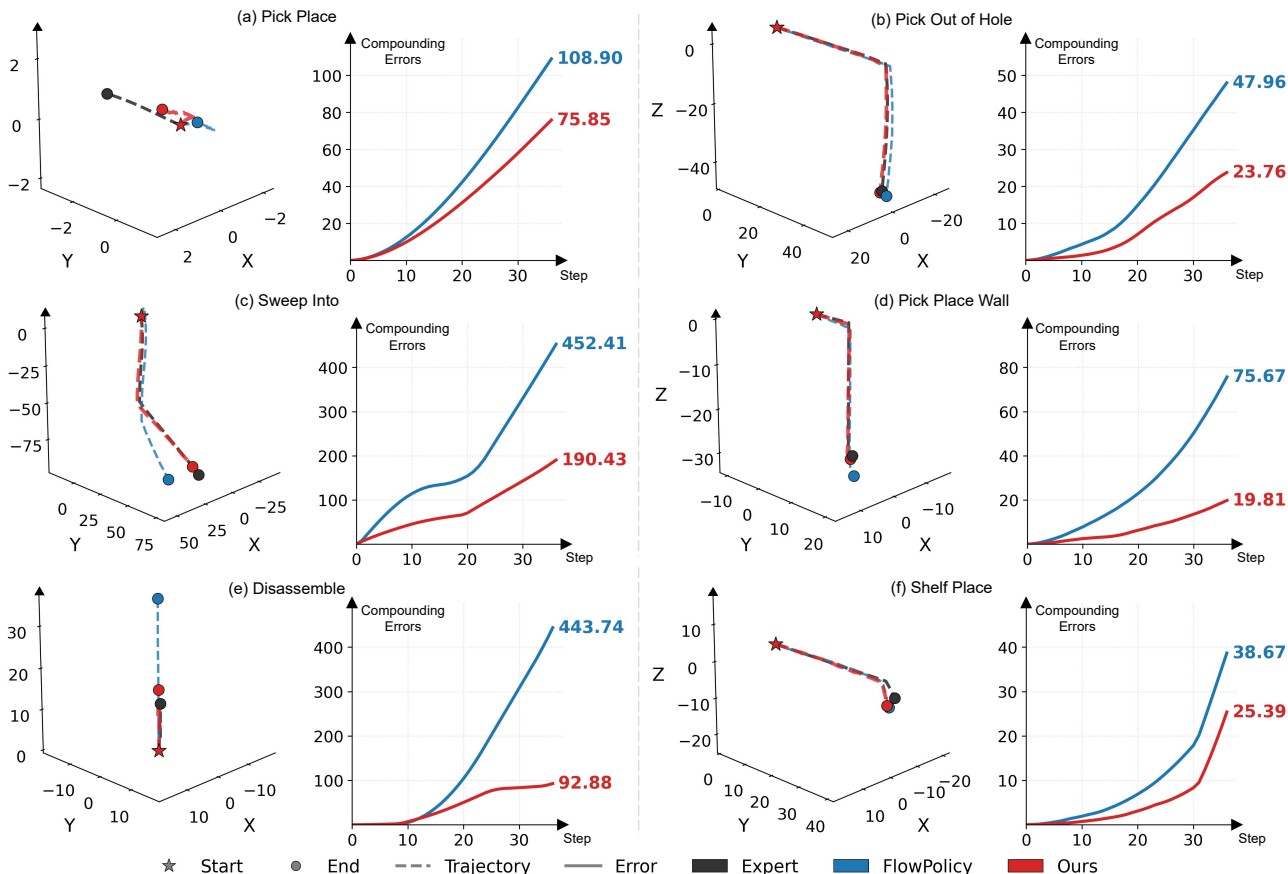

*Figure 9.* Comparison of compounding errors. We visualize the 3D end-effector trajectories (left) inferred by FocalPolicy and FlowPolicy on more representative simulation tasks, given the same initial state. The corresponding error curves (right) represent the *Euclidean distance* between the predicted and ground-truth coordinates ($\|\mathbf{p}_{\mathrm{pred}} - \mathbf{p}_{\mathrm{gt}}\|_2$) at each time step.

context, we utilize TS as a relative measure of imitation fidelity rather than an absolute performance metric. Higher trajectory quality is defined by how closely a policy's TS aligns with the expert's TS, rather than its absolute magnitude. For a more intuitive representation, we define the TS score as $\mathrm{TS}_{\mathrm{score}}(\downarrow) = |\mathrm{TS}_{\mathrm{expert}} - \mathrm{TS}_{\mathrm{policy}}|$ and report these scores in Table 10.

As shown in the table, FocalPolicy achieves lower or equal $\mathrm{TS}_{\mathrm{score}}$ across all 8 evaluated tasks, indicating that its predicted trajectories better match the smoothness of the expert demonstrations. This demonstrates that FocalPolicy significantly improves the modeling of multi-chunk trajectories compared to the baseline FlowPolicy. Through the comparison of these quantitative metrics, we demonstrate that FocalPolicy produces more coherent action trajectories that closely resemble expert actions, effectively addressing the challenge of discontinuity across action chunks in imitation learning. We attribute this improvement to the Foresight Composite Objective (FCO), which enhances the model's foresight in action prediction, and the Locally Anchored Sampling (LAS) strategy, which improves the learning efficiency of the policy.

### C.4. Real-World experiment details

**Hardware Specifications.** Our real-world robotic platform consists of a **UR-10e 6-DOF industrial arm** and an **AG-95 two-finger gripper**. Visual input is captured by a fixed **RealSense D435i RGB-D camera** at a resolution of $840 \times 480$. All computations, including policy inference and vision processing, are performed on a workstation equipped with an **Intel Core i9-14900KF CPU** and an **NVIDIA GeForce RTX 4090 GPU**.

**Expert Demonstration and Training.** For each task, we collected **30-50 expert demonstrations** using a **Logitech F710 Wireless Gamepad**, recording end-effector Cartesian poses and gripper states. During training, we applied standard data augmentations, including random cropping and color jittering, to enhance the policy's visual robustness.

**Task Definitions and Multi-stage Decomposition.** As illustrated in Figure 10, we designed six challenging tasks to evaluate

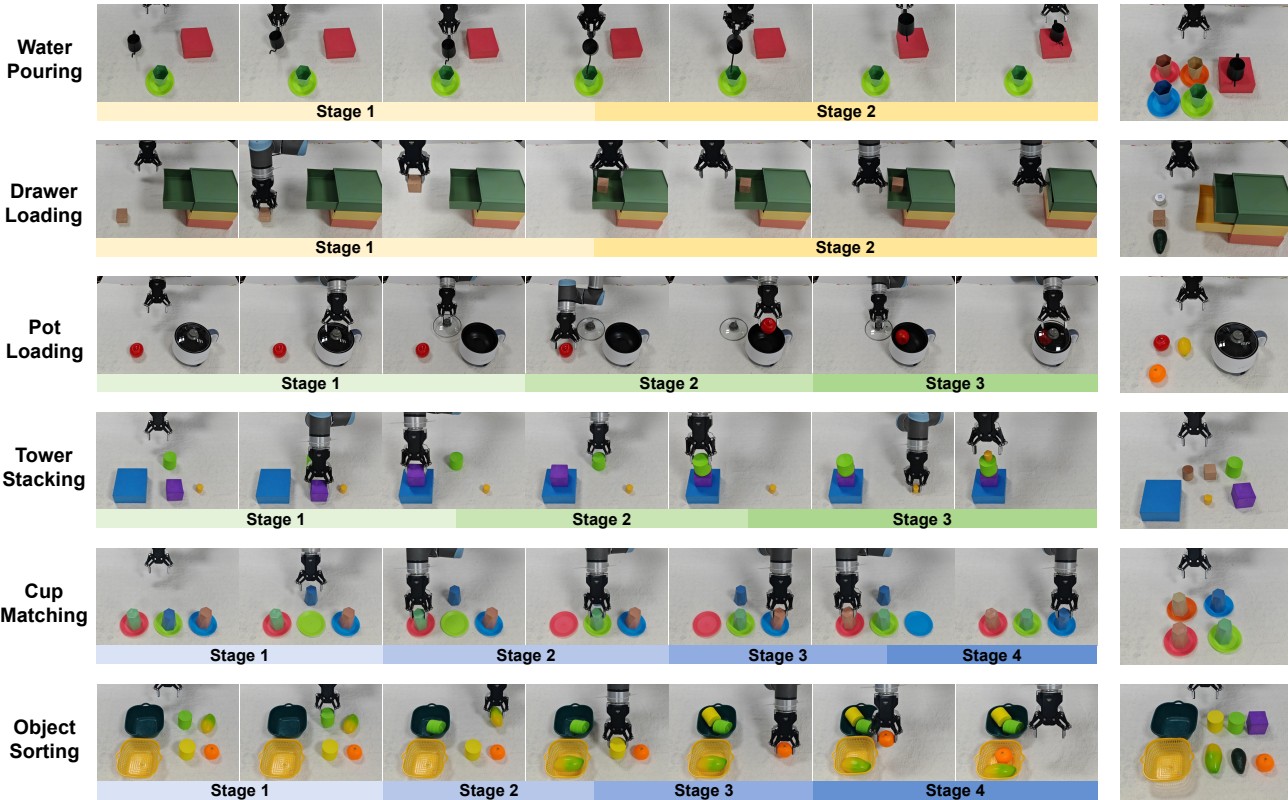

*Figure 10.* Illustration of real-world task phase decomposition. The figure displays the phase segmentation of six tasks on the left, alongside the corresponding variant types for each task on the right.

*Table 11.* Real-world results across staged manipulation tasks. We report **Success Rates** (%) ↑ **/ Score** ↑ for six real-world tasks—*Water Pouring*, *Pot Loading*, *Cup Matching*, *Drawer Loading*, *Tower Stacking*, and *Object Sorting*—each decomposed into sequential stages (Stage 1, Stage 2, *etc.*).

| Alg \ Task | Water Pouring | | Pot Loading | | | Cup Matching | | | |
|---|---|---|---|---|---|---|---|---|---|
| | Stage 1 | Stage 2 | Stage 1 | Stage 2 | Stage 3 | Stage 1 | Stage 2 | Stage 3 | Stage 4 |
| DP3 | 80.0 / 12.0 | 60.0 / 9.0 | 76.7 / 11.5 | 53.3 / 8.0 | 30.0 / 4.5 | 70.0 / 10.5 | 53.3 / 8.0 | 43.3 / 6.5 | 23.3 / 3.5 |
| FreqPolicy | 73.3 / 11.0 | 50.0 / 7.5 | 63.3 / 9.5 | 46.7 / 7.0 | 23.3 / 3.5 | 63.3 / 9.5 | 46.7 / 7.0 | 36.7 / 5.5 | 10.0 / 1.5 |
| FlowPolicy | 83.3 / 12.5 | 53.3 / 8.0 | 66.7 / 10.0 | 50.0 / 7.5 | 20.0 / 3.0 | 60.0 / 9.0 | 50.0 / 7.5 | 33.3 / 5.0 | 13.3 / 2.0 |
| **FocalPolicy** | 86.7 / 13.0 | 63.3 / 9.5 | 76.7 / 11.5 | 53.3 / 8.0 | 33.3 / 5.0 | 73.3 / 11.0 | 56.7 / 8.5 | 50.0 / 7.5 | 30.0 / 4.5 |

| Alg \ Task | Drawer Loading | | Tower Stacking | | | Object Sorting | | | |
|---|---|---|---|---|---|---|---|---|---|
| | Stage 1 | Stage 2 | Stage 1 | Stage 2 | Stage 3 | Stage 1 | Stage 2 | Stage 3 | Stage 4 |
| DP3 | 90.0 / 13.5 | 60.0 / 9.0 | 80.0 / 12.0 | 56.7 / 8.5 | 33.3 / 5.0 | 83.3 / 12.5 | 66.7 / 10.0 | 36.7 / 5.5 | 20.0 / 3.0 |
| FreqPolicy | 86.7 / 13.0 | 53.3 / 8.0 | 76.7 / 11.5 | 53.3 / 8.0 | 23.3 / 3.5 | 86.7 / 13.0 | 70.0 / 10.5 | 33.3 / 5.5 | 16.7 / 2.5 |
| FlowPolicy | 80.0 / 12.0 | 56.7 / 8.5 | 83.3 / 12.5 | 46.7 / 7.0 | 20.0 / 3.0 | 80.0 / 12.0 | 63.3 / 9.5 | 26.7 / 4.0 | 6.7 / 1.0 |
| **FocalPolicy** | 93.3 / 14.0 | 66.7 / 10.0 | 86.7 / 13.0 | 53.3 / 8.0 | 40.0 / 6.0 | 86.7 / 13.0 | 73.3 / 11.0 | 46.7 / 7.0 | 26.7 / 4.0 |

FocalPolicy's ability to handle long-horizon, multi-stage manipulation:

- 1) **Water Pouring** (2 stages, 4 variants): The robot grasps a kettle and maneuvers it above a cup. Subsequently, it pours water into the cup and returns the kettle to a designated resting position. This task challenges the smoothness and accuracy of the manipulation trajectory.

- 2) **Drawer Loading** (2 stages, 6 variants): The robot deposits an object into an open drawer (on either the top or bottom tier) and subsequently closes the drawer using its end-effector. This task evaluates the coordination between free-space object manipulation and articulated mechanism actuation.

- 3) **Pot Loading** (3 stages, 3 variants): The robot lifts and removes the pot lid, deposits an ingredient into the pot, and places the lid back to cover it. This task evaluates the policy's capability to manipulate unattached constraints (the lid) and coordinate sequential interactions within a confined space.

- 4) **Tower Stacking** (3 stages, 3 variants): The robot sequentially stacks three objects of varying shapes and decreasing sizes. This task requires high vertical alignment accuracy and stable object release.

- 5) **Cup Matching** (4 stages, 4 variants): The robot relocates three colored cups onto their corresponding saucers. This task requires strict adherence to color matching, focusing on ordered repetitive operations and precise placement.

- 6) **Object Sorting** (4 stages, 3 variants): The robot sequentially sorts four items into two bins based on attributes. This task involves repeated pick-and-place operations, serving as a benchmark for testing long-horizon execution stability.

**Evaluation Metric.** To quantify performance, we employ a graded scoring system for each stage: $0.5$ points for contacting the target (demonstrating intent) and $1.0$ point for successful completion. We compute the average completion rates over $15$ independent trials per task. Detailed results are presented in Table 11.

## D. Ablation Study Results

This section presents a comprehensive ablation analysis of **FocalPolicy**, accompanied by detailed experimental results. Consistent with the experimental setup in the main manuscript, our comprehensive evaluation spans a total of **24 simulation tasks**. The benchmarks encompass **Adroit** (denoted as **A**) and **MetaWorld** tasks, where the latter are categorized into three difficulty levels based on their complexity: **Medium (M)**, **Hard (H)**, and **Very Hard (VH)**.

The analysis is organized into three parts: First, Sec. D.1 elaborates on the **Component Ablations**, corresponding to the *Component Effectiveness* analysis in the main text. Notably, to gain deeper insights into the contributions of various design choices, we extend our analysis beyond the main manuscript by investigating two supplementary variants: **FCO-Full-Macro** and **FCO-Prox-Freq**. Subsequently, Sec. D.2 presents the **Horizon Ablation**, providing specific configurations and detailed results corresponding to the *Impact of Horizon Hyperparameters* analysis. Third, Appendix D.3 conducts a **Sensitivity Analysis of** $\lambda$, empirically justifying the choice of the balancing coefficient within the Foresight Composite Objective. Finally, Appendix D.4 presents the **Ablation on Spectral Components**, investigating the individual impacts of low-frequency and high-frequency components on modeling long-horizon manipulation trajectories.

### D.1. Component Ablations

To isolate the contributions of the proposed **Foresight Composite Objective (FCO)** and **Locally Anchored Sampling (LAS)**, and to examine the sensitivity of specific design choices, we evaluate a comprehensive set of variants. Unless otherwise specified, all variants maintain the same macro-trajectory horizons $(L, H)$ and training hyperparameters as the original **FocalPolicy** to ensure a controlled comparison. We report success rates (mean $\pm$ std) over three random seeds.

**Ablation Configurations.**  We define the specific implementation details of the compared variants as follows:

- **w/o LAS**: Replaces the locally anchored sampling with standard uniform sampling (as in FlowPolicy), testing the benefit of the biased sampling strategy.

- **w/o FCO & LAS**: Removes both LAS and FCO, yielding a baseline similar to FlowPolicy but using our horizon configuration $(L, H)$.

- $\mathcal{L}_{\text{time}}$**-Only FCO**: Replaces the frequency-domain structural regularization in FCO with a standard time-domain MSE loss acting on the entire macro-trajectory.

- $\mathcal{L}_{\text{freq}}$**-Only FCO**: Trains the policy using solely the frequency-domain loss, removing the temporal alignment constraint on the proximal chunk.

- **Fixed-$r$ LAS**: A deterministic version of LAS where the anchor time $r$ is fixed to the terminal boundary $(r = 1)$. This variant tests the necessity of stochasticity in anchor sampling.

- **FCO-Full-Macro**: Extends the time-domain loss $\mathcal{L}_{\text{time}}$ to the entire macro-trajectory, aligning its horizon with $\mathcal{L}_{\text{freq}}$.

*Table 12.* Ablation study of component effectiveness on **24** tasks. We report the average success rate (%) across tasks.

| Designs | A(3) | M(11) | H(5) | VH(5) | Average |
|---|---|---|---|---|---|
| **Ours** | $\mathbf{76.9 \pm 2.2}$ | $\mathbf{81.9 \pm 1.6}$ | $\mathbf{62.2 \pm 4.6}$ | $\mathbf{85.1 \pm 1.6}$ | **77.9** |
| **w/o LAS** | $71.7 \pm 1.8$ | $65.6 \pm 2.9$ | $47.8 \pm 1.8$ | $73.3 \pm 1.1$ | 64.3 |
| **w/o FCO&LAS** | $46.2 \pm 12.1$ | $56.5 \pm 3.2$ | $43.1 \pm 0.2$ | $64.7 \pm 2.4$ | 54.1 |
| $\mathcal{L}_{\mathbf{time}}$**-Only FCO** | $34.3 \pm 1.2$ | $65.9 \pm 4.8$ | $49.4 \pm 2.1$ | $78.1 \pm 1.7$ | 61.1 |
| $\mathcal{L}_{\mathbf{freq}}$**-Only FCO** | $74.0 \pm 1.0$ | $74.4 \pm 1.9$ | $60.1 \pm 1.3$ | $81.4 \pm 1.8$ | 72.8 |
| **Fixed-$r$ LAS** | $70.4 \pm 2.2$ | $77.4 \pm 2.0$ | $59.0 \pm 3.7$ | $80.2 \pm 1.7$ | 73.3 |
| **FCO-Full-Macro** | $73.8 \pm 0.5$ | $75.4 \pm 1.0$ | $60.2 \pm 1.2$ | $81.9 \pm 1.6$ | 73.4 |
| **FCO-Prox-Freq** | $72.8 \pm 2.5$ | $75.4 \pm 2.1$ | $60.7 \pm 1.0$ | $83.0 \pm 2.7$ | 73.6 |

- **FCO-Prox-Freq**: Reverses our primary design by constraining the frequency-domain loss $\mathcal{L}_{\text{freq}}$ to the proximal chunk ($H$), while applying the time-domain loss $\mathcal{L}_{\text{time}}$ to the full macro-trajectory ($L$).

**Effectiveness of Key Modules.** As shown in Table 12, **FocalPolicy** achieves the strongest and most consistent performance across the Adroit and Meta-World task groups. Removing **LAS** (**w/o LAS**) consistently degrades performance, indicating that locally anchored sampling provides a more effective training signal than uniform time sampling for learning the target distribution. When both modules are removed (**w/o FCO&LAS**), performance drops further, suggesting that **FCO** and **LAS** are complementary: FCO improves trajectory-level coherence via multi-chunk structural regularization, while LAS stabilizes and accelerates optimization by strengthening target-signal propagation.

**Analysis of Objective Designs.** We further analyze the impact of different loss formulations based on the results in Table 12:

- **Time vs. Frequency Supervision:** $\mathcal{L}_{\mathbf{time}}$**-Only FCO** exhibits the most significant performance degradation, particularly on Adroit where the average success rate plummets from 76.9% to 34.3%. This highlights that vanilla temporal alignment is insufficient for capturing the overall motion trend, whereas spectral-domain supervision provides a more structured signal for global trajectory modeling. Conversely, $\mathcal{L}_{\mathbf{freq}}$**-Only FCO** also leads to suboptimal results (e.g., 74.0% on Adroit and 74.4% on Meta-World), indicating that frequency alignment alone lacks the precision required for accurate proximal execution.

- **Horizon Alignment Strategy:** **FCO-Full-Macro** performs generally weaker than FocalPolicy, supporting our design choice of applying the time-domain loss only to the proximal chunk for precision, while using the frequency-domain loss for global trends. Furthermore, the performance drop in **FCO-Prox-Freq** confirms that the primary strength of our DCT-based loss lies in enforcing global structural consistency over the long horizon, rather than serving as a tool for local proximal alignment.

- **Stochasticity in LAS:** The performance of **Fixed-$r$ LAS** is typically inferior to the stochastic version, suggesting that retaining randomness in the anchor time $r$ (while keeping it biased towards the terminal) improves training stability and robustness.

Detailed experimental results for all individual tasks are provided in Table 16.

## D.2. Horizon Ablation.

Table 17 studies the effect of chunk size $H$ and macro multiplier $N$ (i.e., macro-horizon $L = NH$) on Adroit and Meta-World tasks (excluding Easy tasks). Across configurations, moderate macro-horizons (e.g., $H=4$, $N=3$ in our default setting) tend to offer the best trade-off: increasing $N$ enlarges the foresight window and can improve global structure, but overly large $N$ makes the macro-trajectory distribution harder to model and may reduce success rates. Similarly, larger $H$ increases per-chunk prediction difficulty and can hurt performance on tasks that require fine-grained control. These results motivate our default choice of $H=4$ and $N=3$, which provides robust performance across benchmarks while keeping inference efficient.

*Table 13*. Ablation study on the balancing coefficient $\lambda$ across six validation tasks. Results are reported as success rates (0–1).

| $\lambda$ | Bin-Picking | Peg-Insert-Side | Push-Wall | Sweep-Into | Hand-Insert | Shelf-Place | Disassemble | **Average** ↑ |
|---|---|---|---|---|---|---|---|---|
| $1 \cdot 10^{-3}$ | 0.21 | 0.76 | 0.75 | 0.83 | 0.38 | 0.70 | 0.78 | 0.63 |
| $5 \cdot 10^{-4}$ | 0.24 | 0.81 | 0.80 | 0.87 | 0.46 | 0.69 | 0.81 | 0.67 |
| $\mathbf{1 \cdot 10^{-4}}$ | **0.41** | **0.84** | **0.84** | **0.88** | **0.48** | **0.75** | **0.85** | **0.72** |
| $5 \cdot 10^{-5}$ | 0.39 | 0.82 | 0.81 | 0.85 | 0.37 | 0.65 | 0.85 | 0.68 |
| $1 \cdot 10^{-5}$ | 0.15 | 0.74 | 0.38 | 0.24 | 0.10 | 0.54 | 0.76 | 0.42 |

*Table 14*. Ablation study on the impact of low-frequency (LF) and high-frequency (HF) components across four MetaWorld tasks. Results are reported as success rates (0–1).

| Variant | Sweep-Into | Hand-Insert | Pick-Place | Disassemble | **Average** ↑ |
|---|---|---|---|---|---|
| Ours w. Only LF | 0.56 | 0.25 | 0.70 | 0.78 | 0.57 |
| Ours w. Only HF | 0.41 | 0.23 | 0.66 | 0.82 | 0.53 |
| **Ours** | **0.71** | **0.29** | **0.71** | **0.87** | **0.65** |

### D.3. Sensitivity Analysis of $\lambda$.

The hyperparameter $\lambda$ in Eq. (7) balances the proximal time-domain consistency $\mathcal{L}_{\text{time}}$ and the foresight frequency-domain regularization $\mathcal{L}_{\text{freq}}$. As shown in Table 13, we evaluate the sensitivity of FocalPolicy to $\lambda$ by sweeping across five candidates: $\{1 \cdot 10^{-3}, 5 \cdot 10^{-4}, 1 \cdot 10^{-4}, 5 \cdot 10^{-5}, 1 \cdot 10^{-5}\}$ on seven representative validation tasks.

We observe that $\lambda = 10^{-4}$ consistently achieves the highest average success rate of $0.72$. The relatively small magnitude of $\lambda$ is primarily due to the scale difference between the two objectives: $\mathcal{L}_{\text{freq}}$ aggregates spectral information over the entire macro-trajectory $L$, leading to naturally larger numerical gradients compared to the single-chunk $\mathcal{L}_{\text{time}}$. Increasing $\lambda$ to $10^{-3}$ tends to over-regularize the policy, potentially sacrificing the fine-grained precision of proximal actions, while decreasing it to $10^{-5}$ diminishes the foresight-aware structural constraint, leading to a drop in cross-chunk coherence. These empirical results justify our choice of $\lambda = 10^{-4}$ as a robust balancing coefficient.

### D.4. Ablation on Spectral Components.

As discussed in Section 4.1, we posit that low-frequency (LF) components encode the global motion trend of the trajectory, while high-frequency (HF) components represent fine-grained execution details. To empirically validate the necessity of utilizing the full frequency spectrum, we conduct an ablation study isolating the impact of these components.

Table 14 compares the performance of the full FocalPolicy against two degraded variants: one supervised exclusively by the low-frequency component (**Ours w. Only LF**) and another exclusively by the high-frequency component (**Ours w. Only HF**). We evaluate these variants across four representative MetaWorld tasks.

As shown in the results, omitting either spectral component leads to noticeable performance degradation. Specifically, relying solely on LF or HF components results in an average success rate drop of $0.08$ and $0.12$, respectively, compared to our full model ($0.65$). These findings support our hypothesis that the frequency-domain supervision in FocalPolicy effectively captures holistic trajectory patterns by balancing common structural trends (LF) and specific local behaviors (HF). By integrating both, our method provides a comprehensive characterization of global trajectory features across multiple chunks. Exploring more sophisticated mechanisms to dynamically weight or combine these two spectral components remains an interesting direction for future work.

*Table 15.* Complete simulation results across all tasks in Adroit and MetaWorld. Values represent the success rate (mean ± standard deviation) over multiple seeds.

| Alg \ Task | Adroit | | | MetaWorld (Easy) | | | | |
| --- | --- | --- | --- | --- | --- | --- | --- | --- |
| | Hammer | Door | Pen | Button Press | Button Press Topdown | Button Press Topdown Wall | Button Press Wall | Coffee Button |
| DP3 | 98.3 ± 2.9 | 53.5 ± 5.1 | 59.3 ± 3.5 | 100.0 ± 0.0 | 99.7 ± 0.6 | 85.7 ± 8.3 | 100.0 ± 0.0 | 100.0 ± 0.0 |
| FlowPolicy | 94.0 ± 0.0 | 53.0 ± 6.2 | 61.0 ± 3.5 | 100.0 ± 0.0 | 100.0 ± 0.0 | 100.0 ± 0.0 | 100.0 ± 0.0 | 100.0 ± 0.0 |
| FreqPolicy | 94.3 ± 2.1 | 58.3 ± 3.8 | 53.0 ± 3.5 | 100.0 ± 0.0 | 69.3 ± 7.6 | 58.3 ± 2.1 | 100.0 ± 0.0 | 100.0 ± 0.0 |
| **FocalPolicy** | 100.0 ± 0.0 | 63.2 ± 5.3 | 67.5 ± 1.3 | 100.0 ± 0.0 | 100.0 ± 0.0 | 100.0 ± 0.0 | 100.0 ± 0.0 | 100.0 ± 0.0 |

| Alg \ Task | MetaWorld (Easy) | | | | | | | |
| --- | --- | --- | --- | --- | --- | --- | --- | --- |
| | Dial Turn | Door Close | Door Lock | Door Open | Door Unlock | Drawer Close | Drawer Open | Faucet Close |
| DP3 | 65.7 ± 2.3 | 100.0 ± 0.0 | 97.7 ± 0.6 | 100.0 ± 0.0 | 100.0 ± 0.0 | 100.0 ± 0.0 | 100.0 ± 0.0 | 96.0 ± 2.6 |
| FlowPolicy | 87.0 ± 1.0 | 100.0 ± 0.0 | 100.0 ± 0.0 | 100.0 ± 0.0 | 100.0 ± 0.0 | 100.0 ± 0.0 | 100.0 ± 0.0 | 100.0 ± 0.0 |
| FreqPolicy | 65.7 ± 4.2 | 100.0 ± 0.0 | 92.0 ± 6.2 | 100.0 ± 0.0 | 100.0 ± 0.0 | 100.0 ± 0.0 | 98.0 ± 1.7 | 92.7 ± 1.2 |
| **FocalPolicy** | 86.0 ± 1.7 | 100.0 ± 0.0 | 100.0 ± 0.0 | 100.0 ± 0.0 | 100.0 ± 0.0 | 100.0 ± 0.0 | 100.0 ± 0.0 | 100.0 ± 0.0 |

| Alg \ Task | MetaWorld (Easy) | | | | | | | |
| --- | --- | --- | --- | --- | --- | --- | --- | --- |
| | Faucet Open | Handle Press | Handle Pull | Handle Press Side | Handle Pull Side | Lever Pull | Plate Slide | Plate Slide Back |
| DP3 | 100.0 ± 0.0 | 82.7 ± 1.5 | 41.0 ± 9.5 | 100.0 ± 0.0 | 89.3 ± 2.1 | 76.0 ± 5.0 | 100.0 ± 0.0 | 100.0 ± 0.0 |
| FlowPolicy | 100.0 ± 0.0 | 100.0 ± 0.0 | 48.3 ± 21.1 | 100.0 ± 0.0 | 58.0 ± 5.0 | 79.7 ± 2.3 | 100.0 ± 0.0 | 100.0 ± 0.0 |
| FreqPolicy | 100.0 ± 0.0 | 85.0 ± 1.0 | 40.7 ± 10.0 | 100.0 ± 0.0 | 68.3 ± 4.7 | 69.0 ± 5.3 | 100.0 ± 0.0 | 100.0 ± 0.0 |
| **FocalPolicy** | 100.0 ± 0.0 | 100.0 ± 0.0 | 41.3 ± 14.4 | 100.0 ± 0.0 | 56.0 ± 6.2 | 83.3 ± 3.1 | 100.0 ± 0.0 | 100.0 ± 0.0 |

| Alg \ Task | MetaWorld (Easy) | | | | | | |
| --- | --- | --- | --- | --- | --- | --- | --- |
| | Plate Slide Back Side | Plate Slide Side | Reach | Reach Wall | Window Close | Window Open | Peg Unplug Side |
| DP3 | 100.0 ± 0.0 | 100.0 ± 0.0 | 22.7 ± 2.1 | 66.0 ± 5.3 | 100.0 ± 0.0 | 100.0 ± 0.0 | 79.0 ± 3.0 |
| FlowPolicy | 100.0 ± 0.0 | 100.0 ± 0.0 | 27.0 ± 6.1 | 74.7 ± 5.0 | 100.0 ± 0.0 | 100.0 ± 0.0 | 77.3 ± 2.1 |
| FreqPolicy | 100.0 ± 0.0 | 100.0 ± 0.0 | 22.3 ± 3.5 | 74.3 ± 2.5 | 100.0 ± 0.0 | 93.3 ± 0.6 | 62.0 ± 3.6 |
| **FocalPolicy** | 100.0 ± 0.0 | 100.0 ± 0.0 | 24.3 ± 2.9 | 86.3 ± 2.5 | 100.0 ± 0.0 | 100.0 ± 0.0 | 82.3 ± 3.1 |

| Alg \ Task | MetaWorld (Medium) | | | | | | |
| --- | --- | --- | --- | --- | --- | --- | --- |
| | Basketball | Bin Picking | Box Close | Coffee Pull | Coffee Push | Hammer | Peg Insert Side |
| DP3 | 100.0 ± 0.0 | 65.3 ± 14.2 | 59.7 ± 3.2 | 88.3 ± 2.9 | 91.3 ± 6.1 | 100.0 ± 0.0 | 78.7 ± 9.5 |
| FlowPolicy | 88.3 ± 9.8 | 49.7 ± 24.8 | 54.3 ± 6.0 | 95.3 ± 3.2 | 93.0 ± 2.6 | 99.0 ± 1.7 | 83.0 ± 6.1 |
| FreqPolicy | 99.0 ± 1.7 | 28.7 ± 12.4 | 48.0 ± 6.1 | 82.0 ± 7.0 | 87.3 ± 6.7 | 100.0 ± 0.0 | 63.7 ± 9.6 |
| **FocalPolicy** | 100.0 ± 0.0 | 64.3 ± 20.3 | 61.3 ± 1.5 | 98.0 ± 1.0 | 96.7 ± 1.5 | 99.7 ± 0.6 | 88.3 ± 4.5 |

| Alg \ Task | MetaWorld (Medium) | | | | MetaWorld (hard) | | | |
| --- | --- | --- | --- | --- | --- | --- | --- | --- |
| | Push Wall | Soccer | Sweep | Sweep Into | Assembly | Hand Insert | Pick Out of Hole | Pick Place |
| DP3 | 95.3 ± 4.7 | 31.7 ± 8.4 | 99.3 ± 1.2 | 39.3 ± 25.3 | 98.0 ± 1.7 | 14.3 ± 2.9 | 38.3 ± 0.6 | 60.7 ± 11.9 |
| FlowPolicy | 49.0 ± 3.6 | 40.3 ± 8.1 | 100.0 ± 0.0 | 37.3 ± 31.8 | 98.3 ± 1.5 | 16.0 ± 2.6 | 21.0 ± 4.6 | 55.3 ± 2.3 |
| FreqPolicy | 89.7 ± 5.5 | 24.3 ± 1.5 | 90.3 ± 3.1 | 19.3 ± 8.5 | 98.0 ± 1.0 | 16.0 ± 3.0 | 40.0 ± 2.0 | 57.7 ± 4.2 |
| **FocalPolicy** | 81.0 ± 6.6 | 41.0 ± 7.5 | 100.0 ± 0.0 | 70.7 ± 22.0 | 99.7 ± 0.6 | 29.3 ± 16.2 | 35.0 ± 7.2 | 70.7 ± 4.2 |

| Alg \ Task | MetaWorld (Hard) | | MetaWorld (Very Hard) | | | | | Overall Avg |
| --- | --- | --- | --- | --- | --- | --- | --- | --- |
| | Push | Push Back | Shelf Place | Disassemble | Stick Pull | Stick Push | Pick Place Wall | |
| DP3 | 71.3 ± 5.0 | 0.0 ± 0.0 | 58.7 ± 2.1 | 79.3 ± 7.0 | 67.7 ± 5.0 | 100.0 ± 0.0 | 85.0 ± 4.4 | 79.9 |
| FlowPolicy | 80.3 ± 2.5 | 0.0 ± 0.0 | 62.3 ± 6.1 | 71.0 ± 7.8 | 78.3 ± 2.1 | 100.0 ± 0.0 | 77.7 ± 3.2 | 79.4 |
| FreqPolicy | 66.7 ± 10.7 | 0.0 ± 0.0 | 39.3 ± 11.6 | 85.3 ± 1.5 | 72.7 ± 3.8 | 100.0 ± 0.0 | 74.7 ± 2.3 | 75.1 |
| **FocalPolicy** | 76.3 ± 4.7 | 0.0 ± 0.0 | 69.3 ± 4.9 | 86.7 ± 1.5 | 80.7 ± 1.5 | 100.0 ± 0.0 | 89.0 ± 4.4 | 83.6 |

*Table 16.* **Ablation results** on Adroit and MetaWorld (excluding Easy tasks). Values are success rate (%, mean ± std) over multiple seeds.

| Variant \ Task | Adroit | | |
|---|---|---|---|
| | Hammer | Door | Pen |
| **FocalPolicy** | $100.0 \pm 0.0$ | $63.2 \pm 5.3$ | $67.5 \pm 1.3$ |
| **w/o LAS** | $95.0 \pm 2.6$ | $55.0 \pm 2.6$ | $65.0 \pm 5.6$ |
| **w/o FCO&LAS** | $58.0 \pm 8.7$ | $39.0 \pm 3.6$ | $41.7 \pm 29.4$ |
| $\mathcal{L}_{\text{time}}$**-Only FCO** | $55.0 \pm 4.0$ | $7.0 \pm 1.7$ | $41.0 \pm 3.5$ |
| $\mathcal{L}_{\text{freq}}$**-Only FCO** | $97.0 \pm 3.0$ | $57.0 \pm 1.0$ | $68.0 \pm 3.6$ |
| **LAS $\rightarrow r = 1$** | $97.3 \pm 4.6$ | $54.7 \pm 3.5$ | $59.3 \pm 4.0$ |
| **FCO-Full-Macro** | $97.7 \pm 1.5$ | $54.7 \pm 2.5$ | $69.0 \pm 2.0$ |
| **FCO-Prox-Freq** | $99.0 \pm 0.0$ | $56.3 \pm 3.5$ | $63.0 \pm 6.9$ |

| Variant \ Task | MetaWorld (Medium) | | | | | | |
|---|---|---|---|---|---|---|---|
| | Basketball | Bin Picking | Box Close | Coffee Pull | Coffee Push | Hammer | Peg Insert Side |
| **FocalPolicy** | $100.0 \pm 0.0$ | $64.3 \pm 20.3$ | $61.3 \pm 1.5$ | $98.0 \pm 1.0$ | $96.7 \pm 1.5$ | $99.7 \pm 0.6$ | $88.3 \pm 4.5$ |
| **w/o LAS** | $70.0 \pm 9.5$ | $37.0 \pm 18.0$ | $49.7 \pm 10.4$ | $91.7 \pm 2.9$ | $94.7 \pm 3.2$ | $99.7 \pm 0.6$ | $70.3 \pm 11.7$ |
| **w/o FCO&LAS** | $56.3 \pm 20.6$ | $20.0 \pm 7.2$ | $38.0 \pm 10.0$ | $86.7 \pm 5.0$ | $86.0 \pm 2.6$ | $99.0 \pm 1.7$ | $54.3 \pm 7.5$ |
| $\mathcal{L}_{\text{time}}$**-Only FCO** | $84.3 \pm 12.3$ | $50.0 \pm 20.8$ | $34.0 \pm 7.0$ | $93.3 \pm 5.7$ | $85.3 \pm 2.5$ | $98.7 \pm 1.5$ | $76.0 \pm 2.6$ |
| $\mathcal{L}_{\text{freq}}$**-Only FCO** | $94.3 \pm 9.8$ | $51.7 \pm 16.4$ | $49.0 \pm 3.0$ | $98.3 \pm 0.6$ | $96.0 \pm 3.5$ | $100.0 \pm 0.0$ | $76.3 \pm 3.5$ |
| **LAS $\rightarrow r = 1$** | $100.0 \pm 0.0$ | $51.7 \pm 29.2$ | $56.7 \pm 7.5$ | $97.3 \pm 2.3$ | $94.0 \pm 5.0$ | $99.0 \pm 1.7$ | $82.3 \pm 6.8$ |
| **FCO-Full-Macro** | $99.0 \pm 1.7$ | $56.7 \pm 14.5$ | $47.3 \pm 3.8$ | $97.3 \pm 2.3$ | $94.7 \pm 2.9$ | $100.0 \pm 0.0$ | $73.3 \pm 6.4$ |
| **FCO-Prox-Freq** | $99.0 \pm 1.7$ | $54.7 \pm 16.4$ | $49.0 \pm 9.0$ | $96.7 \pm 0.6$ | $93.0 \pm 5.6$ | $100.0 \pm 0.0$ | $74.0 \pm 10.1$ |

| Variant \ Task | MetaWorld (Medium) | | | | MetaWorld (Hard) | | |
|---|---|---|---|---|---|---|---|
| | Push Wall | Soccer | Sweep | Sweep Into | Assembly | Hand Insert | Pick Out Hole |
| **FocalPolicy** | $81.0 \pm 6.6$ | $41.0 \pm 7.5$ | $100.0 \pm 0.0$ | $70.7 \pm 22.0$ | $99.7 \pm 0.6$ | $29.3 \pm 16.2$ | $35.0 \pm 7.2$ |
| **w/o LAS** | $56.7 \pm 4.7$ | $44.7 \pm 12.7$ | $87.0 \pm 9.6$ | $20.3 \pm 1.5$ | $96.7 \pm 2.5$ | $11.0 \pm 1.0$ | $13.3 \pm 6.1$ |
| **w/o FCO&LAS** | $43.3 \pm 10.1$ | $35.7 \pm 7.1$ | $82.7 \pm 4.7$ | $19.3 \pm 5.1$ | $94.3 \pm 3.5$ | $10.3 \pm 2.1$ | $13.3 \pm 4.5$ |
| $\mathcal{L}_{\text{time}}$**-Only FCO** | $46.7 \pm 25.4$ | $38.0 \pm 8.7$ | $77.3 \pm 4.0$ | $41.7 \pm 30.9$ | $93.0 \pm 4.4$ | $23.0 \pm 13.9$ | $6.3 \pm 5.9$ |
| $\mathcal{L}_{\text{freq}}$**-Only FCO** | $51.7 \pm 12.3$ | $35.0 \pm 9.5$ | $95.3 \pm 3.1$ | $70.3 \pm 11.7$ | $100.0 \pm 0.0$ | $22.0 \pm 10.6$ | $33.3 \pm 2.1$ |
| **LAS $\rightarrow r = 1$** | $68.0 \pm 4.4$ | $37.0 \pm 8.7$ | $98.3 \pm 2.9$ | $67.3 \pm 18.6$ | $99.0 \pm 1.7$ | $26.3 \pm 15.2$ | $28.7 \pm 3.1$ |
| **FCO-Full-Macro** | $66.3 \pm 4.0$ | $33.7 \pm 7.8$ | $96.7 \pm 2.9$ | $64.0 \pm 17.1$ | $99.7 \pm 0.6$ | $21.3 \pm 8.4$ | $34.0 \pm 7.0$ |
| **FCO-Prox-Freq** | $67.3 \pm 12.1$ | $35.7 \pm 13.7$ | $98.3 \pm 1.5$ | $61.3 \pm 21.0$ | $99.7 \pm 0.6$ | $22.0 \pm 8.7$ | $34.7 \pm 3.1$ |

| Variant \ Task | MetaWorld (Hard) | | MetaWorld (Very Hard) | | | | |
|---|---|---|---|---|---|---|---|
| | Pick Place | Push | Shelf Place | Disassemble | Stick Pull | Stick Push | Pick Place Wall |
| **FocalPolicy** | $70.7 \pm 4.2$ | $76.3 \pm 4.7$ | $69.3 \pm 4.9$ | $86.7 \pm 1.5$ | $80.7 \pm 1.5$ | $100.0 \pm 0.0$ | $89.0 \pm 4.4$ |
| **w/o LAS** | $63.0 \pm 5.3$ | $55.0 \pm 4.6$ | $52.3 \pm 5.5$ | $67.3 \pm 2.9$ | $68.3 \pm 4.0$ | $100.0 \pm 0.0$ | $78.7 \pm 4.5$ |
| **w/o FCO&LAS** | $56.0 \pm 3.5$ | $41.7 \pm 3.5$ | $41.7 \pm 2.1$ | $60.7 \pm 4.5$ | $53.3 \pm 5.1$ | $99.7 \pm 0.6$ | $68.0 \pm 7.9$ |
| $\mathcal{L}_{\text{time}}$**-Only FCO** | $67.3 \pm 7.6$ | $57.3 \pm 1.5$ | $57.0 \pm 3.0$ | $83.0 \pm 1.7$ | $79.7 \pm 2.5$ | $100.0 \pm 0.0$ | $70.7 \pm 10.0$ |
| $\mathcal{L}_{\text{freq}}$**-Only FCO** | $67.0 \pm 9.5$ | $78.0 \pm 7.0$ | $67.7 \pm 6.1$ | $79.7 \pm 2.1$ | $77.7 \pm 2.5$ | $100.0 \pm 0.0$ | $82.0 \pm 9.0$ |
| **LAS $\rightarrow r = 1$** | $67.0 \pm 6.6$ | $74.0 \pm 6.1$ | $56.3 \pm 4.5$ | $82.0 \pm 5.6$ | $76.7 \pm 3.2$ | $100.0 \pm 0.0$ | $86.0 \pm 10.4$ |
| **FCO-Full-Macro** | $68.3 \pm 7.0$ | $77.7 \pm 6.4$ | $69.0 \pm 7.2$ | $80.3 \pm 3.8$ | $76.7 \pm 4.7$ | $100.0 \pm 0.0$ | $83.7 \pm 6.0$ |
| **FCO-Prox-Freq** | $68.0 \pm 6.2$ | $79.0 \pm 1.7$ | $70.0 \pm 3.0$ | $80.3 \pm 2.5$ | $80.7 \pm 5.5$ | $100.0 \pm 0.0$ | $84.0 \pm 5.3$ |

*Table 17.* **Horizon ablation results** on Adroit and MetaWorld (excluding Easy tasks). Values are success rate (%, mean ± std) over multiple seeds.

| H | N | Adroit | | |
|---|---|---|---|---|
| | | Hammer | Door | Pen |
| 4 | 2 | 97.7 ± 2.1 | 59.7 ± 6.7 | 63.3 ± 2.5 |
| | **3** | **100.0 ± 0.0** | **63.2 ± 5.3** | **67.5 ± 1.3** |
| | 4 | 98.3 ± 2.1 | 61.3 ± 5.9 | 68.7 ± 4.5 |
| | 5 | 95.0 ± 4.4 | 56.7 ± 4.2 | 59.7 ± 9.1 |
| 8 | 2 | 97.7 ± 1.5 | 61.3 ± 0.6 | 62.0 ± 7.0 |
| | 3 | 97.3 ± 2.5 | 61.7 ± 4.7 | 64.3 ± 4.5 |
| | 4 | 98.0 ± 1.7 | 63.0 ± 6.0 | 59.7 ± 5.6 |
| | 5 | 93.7 ± 1.5 | 55.7 ± 8.6 | 68.3 ± 6.1 |
| 12 | 2 | 97.3 ± 3.1 | 54.3 ± 2.1 | 66.3 ± 1.5 |
| | 3 | 94.7 ± 4.0 | 56.7 ± 2.3 | 69.3 ± 2.1 |
| | 4 | 93.3 ± 4.7 | 48.0 ± 3.6 | 67.7 ± 2.1 |
| | 5 | 91.0 ± 4.4 | 52.0 ± 0.0 | 65.3 ± 2.3 |

| H | N | MetaWorld (Medium) | | | | | | |
|---|---|---|---|---|---|---|---|---|
| | | Basketball | Bin Picking | Box Close | Coffee Pull | Coffee Push | Hammer | Peg Insert Side |
| 4 | 2 | 99.0 ± 1.7 | 55.0 ± 26.0 | 57.7 ± 11.9 | 95.3 ± 2.5 | 92.3 ± 3.2 | 99.0 ± 1.7 | 84.0 ± 7.0 |
| | **3** | **100.0 ± 0.0** | **64.3 ± 20.3** | **61.3 ± 1.5** | **98.0 ± 1.0** | **96.7 ± 1.5** | **99.7 ± 0.6** | **88.3 ± 4.5** |
| | 4 | 99.7 ± 0.6 | 55.0 ± 25.0 | 57.0 ± 2.6 | 96.3 ± 1.2 | 93.3 ± 3.2 | 99.0 ± 1.7 | 85.0 ± 4.4 |
| | 5 | 83.3 ± 11.9 | 15.3 ± 7.5 | 48.0 ± 1.0 | 90.0 ± 3.5 | 93.0 ± 1.7 | 99.0 ± 1.0 | 78.0 ± 1.0 |
| 8 | 2 | 99.0 ± 1.7 | 69.7 ± 13.8 | 53.7 ± 9.5 | 95.7 ± 2.1 | 92.7 ± 2.5 | 100.0 ± 0.0 | 80.7 ± 11.0 |
| | 3 | 92.7 ± 4.9 | 40.7 ± 36.7 | 51.0 ± 7.8 | 95.3 ± 1.5 | 94.0 ± 3.6 | 99.0 ± 1.7 | 78.7 ± 3.5 |
| | 4 | 100.0 ± 0.0 | 66.3 ± 11.6 | 49.7 ± 7.4 | 90.7 ± 3.1 | 93.3 ± 5.5 | 99.3 ± 1.2 | 80.7 ± 9.5 |
| | 5 | 99.7 ± 0.6 | 57.7 ± 23.2 | 44.3 ± 4.2 | 92.7 ± 0.6 | 94.3 ± 4.0 | 100.0 ± 0.0 | 82.0 ± 7.5 |
| 12 | 2 | 96.3 ± 2.5 | 63.7 ± 14.6 | 46.3 ± 5.5 | 91.3 ± 1.2 | 90.3 ± 4.0 | 100.0 ± 0.0 | 72.0 ± 9.8 |
| | 3 | 99.7 ± 0.6 | 68.7 ± 12.5 | 49.7 ± 2.1 | 90.7 ± 0.6 | 91.3 ± 6.4 | 100.0 ± 0.0 | 74.0 ± 7.2 |
| | 4 | 97.7 ± 2.5 | 60.7 ± 14.3 | 42.3 ± 2.5 | 88.7 ± 3.5 | 91.0 ± 5.2 | 100.0 ± 0.0 | 69.7 ± 10.1 |
| | 5 | 96.0 ± 4.0 | 61.3 ± 12.9 | 38.7 ± 7.6 | 88.7 ± 2.5 | 90.0 ± 4.6 | 100.0 ± 0.0 | 70.3 ± 10.6 |

| H | N | MetaWorld (Medium) | | | | MetaWorld (Hard) | | |
|---|---|---|---|---|---|---|---|---|
| | | Push Wall | Soccer | Sweep | Sweep Into | Assembly | Hand Insert | Pick Out Hole |
| 4 | 2 | 48.3 ± 12.6 | 37.7 ± 8.5 | 98.7 ± 1.2 | 54.7 ± 21.8 | 98.0 ± 3.5 | 25.3 ± 14.5 | 37.7 ± 3.8 |
| | **3** | **81.0 ± 6.6** | **41.0 ± 7.5** | **100.0 ± 0.0** | **70.7 ± 22.0** | **99.7 ± 0.6** | **29.3 ± 16.2** | **35.0 ± 7.2** |
| | 4 | 74.0 ± 9.8 | 37.7 ± 9.7 | 99.7 ± 0.6 | 45.0 ± 36.8 | 98.3 ± 2.1 | 24.3 ± 17.1 | 29.7 ± 4.5 |
| | 5 | 44.7 ± 8.1 | 44.0 ± 11.5 | 96.7 ± 3.2 | 33.0 ± 36.6 | 93.7 ± 2.1 | 12.7 ± 2.1 | 16.7 ± 6.5 |
| 8 | 2 | 74.7 ± 5.9 | 40.7 ± 9.0 | 97.3 ± 3.8 | 60.0 ± 13.2 | 97.7 ± 2.5 | 21.7 ± 9.0 | 29.3 ± 2.1 |
| | 3 | 54.0 ± 7.9 | 42.0 ± 9.2 | 94.7 ± 0.6 | 41.7 ± 23.7 | 99.3 ± 1.2 | 15.3 ± 2.1 | 20.0 ± 4.6 |
| | 4 | 79.3 ± 4.5 | 37.3 ± 11.0 | 98.0 ± 2.6 | 58.0 ± 24.2 | 98.3 ± 1.5 | 23.0 ± 7.0 | 33.3 ± 3.1 |
| | 5 | 82.0 ± 1.0 | 37.3 ± 7.8 | 97.7 ± 4.0 | 41.7 ± 38.2 | 93.7 ± 1.2 | 24.0 ± 10.6 | 32.0 ± 6.0 |
| 12 | 2 | 76.0 ± 6.2 | 35.7 ± 11.1 | 91.7 ± 3.5 | 49.3 ± 21.4 | 99.3 ± 1.2 | 16.7 ± 5.5 | 35.0 ± 2.6 |
| | 3 | 79.3 ± 2.9 | 36.0 ± 5.6 | 93.3 ± 2.3 | 61.3 ± 25.3 | 98.7 ± 1.5 | 16.7 ± 9.0 | 38.0 ± 5.2 |
| | 4 | 77.0 ± 11.3 | 34.7 ± 9.5 | 91.0 ± 3.6 | 49.3 ± 24.8 | 99.0 ± 1.7 | 14.3 ± 2.1 | 33.7 ± 6.4 |
| | 5 | 72.7 ± 15.0 | 35.7 ± 8.1 | 91.0 ± 5.6 | 45.3 ± 30.0 | 99.3 ± 1.2 | 18.0 ± 7.9 | 38.7 ± 1.2 |

| H | N | MetaWorld (Hard) | | | MetaWorld (Very Hard) | | | |
|---|---|---|---|---|---|---|---|---|
| | | Pick Place | Push | Shelf Place | Disassemble | Stick Pull | Stick Push | Pick Place Wall |
| 4 | 2 | 60.0 ± 1.0 | 74.3 ± 8.1 | 68.3 ± 4.5 | 80.3 ± 3.8 | 78.7 ± 4.2 | 100.0 ± 0.0 | 84.3 ± 3.1 |
| | **3** | **70.7 ± 4.2** | **76.3 ± 4.7** | **69.3 ± 4.9** | **86.7 ± 1.5** | **80.7 ± 1.5** | **100.0 ± 0.0** | **89.0 ± 4.4** |
| | 4 | 64.3 ± 5.0 | 76.7 ± 3.1 | 67.3 ± 6.5 | 82.7 ± 3.1 | 78.0 ± 6.6 | 100.0 ± 0.0 | 82.3 ± 8.6 |
| | 5 | 56.0 ± 1.0 | 68.7 ± 4.5 | 49.0 ± 2.6 | 71.3 ± 5.1 | 72.3 ± 2.1 | 100.0 ± 0.0 | 74.3 ± 2.5 |
| 8 | 2 | 72.3 ± 3.8 | 76.0 ± 1.7 | 60.7 ± 5.1 | 79.7 ± 3.8 | 75.7 ± 3.1 | 100.0 ± 0.0 | 85.7 ± 4.7 |
| | 3 | 70.3 ± 3.8 | 80.3 ± 1.5 | 58.3 ± 4.9 | 78.7 ± 5.7 | 73.3 ± 4.7 | 100.0 ± 0.0 | 84.3 ± 3.2 |
| | 4 | 68.0 ± 5.0 | 77.3 ± 4.0 | 61.3 ± 4.5 | 85.0 ± 3.6 | 75.7 ± 2.1 | 100.0 ± 0.0 | 88.0 ± 1.7 |
| | 5 | 65.7 ± 5.9 | 77.7 ± 2.9 | 60.3 ± 8.5 | 81.7 ± 2.5 | 73.0 ± 4.6 | 100.0 ± 0.0 | 85.0 ± 2.6 |
| 12 | 2 | 75.3 ± 6.4 | 80.7 ± 4.2 | 58.3 ± 4.9 | 82.7 ± 1.5 | 72.0 ± 6.1 | 100.0 ± 0.0 | 83.3 ± 4.9 |
| | 3 | 73.0 ± 6.2 | 81.7 ± 5.8 | 57.7 ± 3.8 | 84.0 ± 3.6 | 72.7 ± 3.1 | 100.0 ± 0.0 | 82.3 ± 4.5 |
| | 4 | 71.0 ± 5.0 | 81.0 ± 5.3 | 56.0 ± 7.9 | 81.7 ± 4.0 | 67.3 ± 3.5 | 100.0 ± 0.0 | 81.3 ± 3.2 |
| | 5 | 71.7 ± 5.8 | 79.0 ± 5.2 | 52.3 ± 11.6 | 76.7 ± 4.5 | 68.3 ± 1.5 | 100.0 ± 0.0 | 84.0 ± 3.5 |

