# OpenReview forum: "FocalPolicy: Frequency-Optimized Chunking and Locally Anchored Flow Matching for Coherent Visuomotor Policy"
_ICML.cc/2026/Conference — ICML 2026 regular_

### Official Review · Reviewer_f7R3 · 2026-03-10

**Soundness:** 3
**Presentation:** 3
**Significance:** 3
**Originality:** 2
**Overall Recommendation:** 5
**Confidence:** 3

**Summary:**

This paper proposes FocalFocalPolicy, the core is Foresight Composite Objective, which uses both frequency and time domain supervision signals for action chunk coherence. It also proposes  Locally Anchored flow matching for more efficient training,

**Compliance With Llm Reviewing Policy:**

Affirmed.

**Final Justification:**

My concerns have been adequately addressed. I think this is a solid paper and should  be accepted.

**Key Questions For Authors:**

1. What are the number of function evaluations for FOCAL policy?
2. You mentioned L_time when introducing L_freq. It seems a bit confusing, as L_time was not introduced at that part.  Maybe it could be better organized? Also, "In FCO,
where the Ltime supervises only single-chunk, ", but it seems there is no L_time in FCO objective, and this makes me confused.
3. It seems FCO objective is not formally defined？ I could be missing something, but this really confuses me. Is FCO the same as L_freq? which equation corresponds to FCO?
4. Does L_time apply to all action chunks? Is that right? Then actually you are supervising both proximal and future action chunks with
both frequency and time domain signals? Because Mt by equation (4) contains all action chunks? This seems contradicts with your description? This also makes me confused with the design choice, why the L_time signal could be used for future action chunks, which should be helpful or why it is not helpful?
5. How do you choose $\lambda$ to balance L_freq and L_time? Why the weight L_freq seems to be much lower?
6. "For each seed, we evaluate the policy 20 times every 200 epochs and report the average of the top five success
rates,", what do you mean by average of the top five success rates? And why do you use this metric?

**Limitations:**

yes

**Strengths And Weaknesses:**

Strength:
1. The idea is interesting and problem being solved has high impact on robotics.
2. Experiments are extensive and strong. Ablation is thorough. Real-world robot tasks further demonstrates effectiveness.
3. There are lots of efforts in writing, and the appendix looks good.


Weakness:
1. It seems the results come from 3 random seeds. Is it a bit small? Adding more seeds could be helpful.
2. What is new in the frequency domain supervision compared to prior work working on frequency domain? does the main novelty here come from applying this supervision to future action chunks?
3. The soft masking and inpainting strategy proposed in RTC, why is it intra-chunk? Isn't it focusing on inter-chunk discontinuities? If so, why not consider comparing with RTC? Or at least add some discussions on this point?

---

> ### Author Rebuttal · Authors · 2026-03-26
>
> We thank you for recognizing the contributions of our work, including the idea, writing and experiments.
> > __W1:__ Concern on the number of random seeds.
>
> __A1:__ Following prior work [1][2][3], we use three seeds, which is common in robot learning community. Following your advice, we evaluate the four MetaWorld tasks in Table 2 with five seeds. In new experiments, FocalPolicy's variance slightly increases from ±0.11 to ±0.12, while it outperforms FlowPolicy by 0.16. We will include these results in the revised paper to demonstrates the stability of FocalPolicy.
> > __W2:__ Question about the novelty of frequency-domain supervision.
>
> __A2:__ Our main innovation is to augment proximal-time supervision with frequency-domain supervision over longer action chunks, and we demonstrate that this combination helps the model overcome incoherence across action chunks. Furthermore, the proposed modules, FCO and LAS, are designed as plug-and-play components that can be readily integrated to improve similar flow- and diffusion-based policies.
> > __W3:__ Comment on RTC and compare with FocalPolicy.
>
> __A3.1:__ Thank you for the question, and we apologize for the unclear phrasing. Our citations of RTC (L35, 2nd col; L86, 1st col) are intended to corroborate the critical challenge of inter-chunk incoherence in robot manipulation, rather than to classify RTC as an intra-chunk work.
>
> __A3.2:__ Our method differs fundamentally from RTC in whether it participates in model training. To mitigate action-chunk incoherence, we optimize the training objective of the foundation model, whereas RTC improves the inference process of a trained VLA without involving training. Considering the difference in parameter scale, we do not compare FocalPolicy ($\approx$255M) with RTC-based VLA models ($>$1B). We have added relevant discussion in the revised paper and will explore the performance of FocalPolicy in VLA in future work.
> > __Q1:__ The Number of Function Evaluations (NFE) in FocalPolicy.
>
> __A1:__ Our NFE is 1 and noted in Table 1; second column.
> > __Q2&3:__ Confusion about FCO, $L_{time}$ and $L_{freq}$ .
>
> __A2&3:__ We apologize for the confusion and thank you for pointing this out. We clarify that FCO includes both the single-chunk $L_{time}$ and the multi-chunk $L_{freq}$, as shown in Fig. 2 (right). In fact, the formal definition of FCO corresponds to Eq. (14) in Section 4.3; that means $L_{FCO}\equiv L_{total}$.
> To improve clarity, we have made the following revisions:
> + Replace $L_{total}$ with $L_{FCO}$, and move the discussion in Section 4.3 to Section 4.1 for a clearer definition of FCO.
> + Add an explanation of $L_{time}$ when it is first introduced in Section 4.1, together with a reference to Eq. (13), to make its relation to FCO and $L_{freq}$ more explicit.
>
> > __Q4:__ Discussion on the temporal length of $L_{time}$ and the choice of short-time supervision.
>
> __A4:__ Thank you for highlighting the ambiguity regarding $L_{time}$. We respond to the two key concerns as follows:
> + Scope of $L_{time}$. $L_{time}$ is applied only to the proximal single chunk ($1:H$), rather than all chunks. This is consistent with Eq. (13), Fig. 2, and our discussions (e.g., L69-72, L190-193). We apologize for the notation ambiguity (e.g., $M_1\equiv M_t$ in L254) and have explicitly clarified this in the revised paper.
> + Design of $L_{time}$. Prior work [2] shows that $L_{time}$ is effective for fine-grained alignment, and $L_{freq}$ better captures global spatio-temporal correlations. In FCO, we leverage this complementarity to learn both precise and coherent manipulation. As shown in Table 9 in paper, the variant (FCO-Full-Macro) applying $L_{time}$ and $L_{freq}$ over the same length underperforms on 24 tasks.
>
> > __Q5:__ Choosing the weight $\lambda$.
>
> __A5:__ We evaluate $\lambda=1e-3,5e-4,1e-4,5e-5,1e-5$ on six validation tasks. $\lambda=1e-4$ yields the best average success rate, so we adopt it in FocalPolicy.
> $\lambda$|1e-3|5e-4|1e-4|5e-5|1e-5
> ---|---|---|---|---|---
> Bin-Picking|0.21|0.24|0.41|0.39|0.15
> Peg-Insert-Side|0.76|0.81|0.84|0.82|0.74
> Push-Wall|0.75|0.80|0.84|0.81|0.38
> Sweep-Into|0.83|0.87|0.88|0.85|0.24
> Hand-Insert|0.38|0.46|0.48|0.37|0.10
> Shelf-Place|0.70|0.69|0.75|0.65|0.54
> Disassemble|0.78|0.81|0.85|0.85|0.76
> Average$\uparrow$|0.63|0.67|0.72|0.68|0.42
> > __Q6:__ Clarification of the "average of top five success rates".
>
> __A6:__ For each seed, we conduct 15 evaluations during a 3000-epoch run (every 200 epochs), with each evaluation averaged over 20 test episodes. We then average the highest 5 results from these 15 evaluations. As noted in W&A1, this metric is common in robot learning community [1][2][3].
>
> [1] 3D Diffusion Policy: Generalizable Visuomotor Policy Learning via Simple 3D Representations, RSS 2024
> [2] FreqPolicy: Frequency Autoregressive Visuomotor Policy with Continuous Tokens, NeurIPS 2025
> [3] MP1: MeanFlow Tames Policy Learning in 1-step for Robotic Manipulation, AAAI 2026

---

> > ### Author Rebuttal · Reviewer_f7R3 · 2026-04-01
> >
> > I would increase my score to 5.

---

> > > ### Author Response · Authors · 2026-04-01
> > >
> > > Thank you again for your time and for these very helpful suggestions that have substantially improved the paper. We are glad to hear that our previous responses addressed most of your concerns.

---

### Official Review · Reviewer_EWJo · 2026-03-13

**Soundness:** 3
**Presentation:** 4
**Significance:** 3
**Originality:** 3
**Overall Recommendation:** 5
**Confidence:** 4

**Summary:**

This paper aims to address the impact of inter-chunk discontinuities on long-horizon actions, and proposes FocalPolicy, a foresight-aware visuomotor policy that combines frequency-optimized chunking with locally anchored flow matching. Specifically, it introduces a foresight composite objective to integrate proximal time-domain alignment with multi-chunk frequency-domain structural regularization. The proposed method has been verified for its effectiveness both in the simulation environment and in the real environment.

**Compliance With Llm Reviewing Policy:**

Affirmed.

**Key Questions For Authors:**

I'm wondering if there is a quantitative metric to measure inter-chunk discontinuities.

**Limitations:**

Yes.

**Strengths And Weaknesses:**

Strengths:

1. This paper presents a new method to address the inter-chunk discontinuity issue in long-horizon actions.
2. This paper is written clearly and fluently, with strong readability.
3. This paper conducts sufficient comparative experiments and ablation studies.

Weaknesses:

In Section 4.1 of the paper, it is mentioned that low-frequency components encode the global motion trend, while high-frequency components represent fine-grained details. Could some related ablation studies be conducted to see the impact of only low-frequency components or only high-frequency components on performance?

---

> ### Author Rebuttal · Authors · 2026-03-26
>
> We thank the reviewer for recognizing our contributions, particularly noting that our paper (1) presents a clear and (2) well-structured approach to addressing inter-chunk incoherence, (3) supported by sufficient comparative experiments and ablation studies.
>
> > __W1:__ Questions regarding the ablation study on isolated low-frequency and high-frequency components.
>
> __A1:__ We thank the reviewer for this insightful concern. We have conducted additional ablation experiments regarding the high-frequency and low-frequency components across the four MetaWorld tasks in Table 2. As shown in the table below, models using only the high-frequency component (Ours w. Only HF) and only the low-frequency component (Ours w. Only LF) achieve average success rates that drop by 0.12 and 0.08, respectively, compared to our FocalPolicy. We will include these experimental results in the revised version, which makes our work more comprehensive and solid.
> Task|Ours w. Only LF|Ours w. Only HF|Ours
> ---|---|---|---
> Sweep-Into|0.56|0.41|0.71
> Hand-Insert|0.25|0.23|0.29
> Pick-Place|0.70|0.66|0.71
> Disassemble|0.78|0.82|0.87
> Average$\uparrow$|0.57|0.53|0.65
>
> We hypothesize that frequency-domain supervision in FocalPolicy captures global trajectory patterns across multiple action chunks, balancing common (low-frequency) and specific (high-frequency) behaviors. To provide a balanced characterization of global trajectory features across multiple chunks, we do not explicitly favor either low-frequency or high-frequency components. We believe that how to better combine the two for trajectory representation could be a meaningful direction for future investigation.
>
> > __Q1:__ Inquiry about quantitative metrics to measure inter-chunk discontinuities.
>
> __A1:__ Thank you for raising this concern. Following recent studies, we select Action Total Variation (ATV $\downarrow$) in ACG [1] to quantify the temporal coherence of the action sequences. The definition of ATV is $\mathrm{ATV}=\frac{1}{M(T-1)}\sum_{t=1}^{T-1}\sum_{j=1}^{M}|a_{t+1}^{j}-a_{t}^{j}|$, where $t$ denotes the time step in the trajectory and $j$ denotes the action dimension. We compute the ATV values of the trajectories in Tables 7 and 9 in the paper and report them in the table below. As shown in the table below, FocalPolicy achieves lower ATV scores than FlowPolicy across all tasks, indicating that our model effectively alleviates discontinuities across action chunks.
> ATV$\downarrow$|FlowPolicy|Ours
> ---|---|---
> Bin Picking|0.99|0.87
> Disassemble|0.35|0.21
> Pick Out of Hole|0.94|0.92
> Pick Place|0.03|0.03
> Pick Place Wall|0.42|0.39
> Push Wall|0.39|0.33
> Shelf place|0.48|0.46
> Sweep Into|1.66|1.45
>
> Furthermore, we calculated the Trajectory Smoothness ($TS$) metric in [2] to evaluate the smoothness of the predicted trajectories. In this comparison, we use $TS$ as a relative measure of imitation fidelity rather than an absolute performance metric. Higher trajectory quality is defined by how closely a policy’s $TS$ aligns with the expert’s $TS$, rather than its absolute magnitude. For a more intuitive representation, we define $TS_{score} (\downarrow) = |TS_{expert}-TS_{policy}|$ and report these scores in the table below. FocalPolicy achieves lower $TS_{score}$ across 8 tasks, indicating that its predicted trajectories better match the smoothness of expert trajectories. This demonstrates that FocalPolicy improves the modeling of multi-chunk trajectories compared to the baseline FlowPolicy.
> $TS_{score}\downarrow$|FlowPolicy|Ours
> ---|---|---
> Bin Picking|0.27|0.13
> Disassemble|0.09|0.01
> Pick Out of Hole|0.04|0.02
> Pick Place|0.01|0.01
> Pick Place Wall|0.03|0.00
> Push Wall|0.09|0.00
> Shelf place|0.03|0.01
> Sweep Into|0.50|0.40
>
> Through the comparison of the above quantitative metrics, we demonstrate that FocalPolicy produces more coherent action trajectories that are closer to expert actions, effectively addressing the challenge of discontinuity across action chunks in imitation learning policies. We attribute this improvement to FCO, which enhances the model’s foresight in action prediction, and LAS, which improves the learning efficiency of the policy. We appreciate your constructive suggestion and have included these quantitative results in our paper.
>
> [1] ACG: Action Coherence Guidance for Flow-based VLA models, ICRA 26
> [2] CoLA-Flow Policy: Temporally Coherent Imitation Learning via Continuous Latent Action Flow Matching for Robotic Manipulation, arXiv 26

---

> > ### Author Rebuttal · Reviewer_EWJo · 2026-04-03
> >
> > Since all my questions and weaknesses have been satisfactorily resolved, I maintain my positive evaluation.

---

> > > ### Author Response · Authors · 2026-04-04
> > >
> > > We would like to thank you again for taking the time to carefully read our work and for recognizing our contributions. Your discussion of the frequency-domain component ablation experiments and the action coherence metric greatly helps validate our work.

---

### Official Review · Reviewer_1ToN · 2026-03-13

**Soundness:** 4
**Presentation:** 3
**Significance:** 4
**Originality:** 2
**Overall Recommendation:** 5
**Confidence:** 4

**Summary:**

FocalPolicy tackles the problem on inter-chunk inconsistencies that hurdles chunk-based flow-matching trajectory prediction approaches in robotics. Extending FlowPolicy, it introduces the Foresight Composite Objective (FCO), which combines proximal time-domain action refinement with distal frequency-domain regularization to resolve inter-chunk discontinuities. Locally Anchored Sampling (LAS), which concentrates supervision near chunk terminals (almost expert-like) to improve robustness and alignment.

**Compliance With Llm Reviewing Policy:**

Affirmed.

**Final Justification:**

I believe the work provides value, and I will maintain my score.

**Key Questions For Authors:**

Questions:
- Is my understanding correct that, in the paper’s default setting, the “proximal” section refers only to the first action chunk of length H, so with H=4 and N=3 the proximal section is 4 steps, while the full macro-trajectory spans NH=12 steps, even though the underlying task trajectory is much longer than 12 steps?
- In the compounding-error analysis, my understanding is that FocalPolicy uses its default multi-chunk setup (N=3, H=4). FlowPolicy does not have an analogous N; but what is the value of H used by FlowPolicy? Even if they do not match, it would be valuable to have the value appear in the text/caption of Figure 7.

**Limitations:**

yes

**Strengths And Weaknesses:**

Strengths:
- The idea is simple and scales well.
- Good balancing and repartition of context across intro+related work.
- Massive amount of environments in experimental setup (incl. 6 real-world long-horizon): good.
- I appreciate that the paper does not leave the interaction between the proximal time-domain loss and the full-macro frequency-domain loss implicit, but instead explicitly argues that their gradients are non-opposing on the proximal chunk. The analysis of spectral sensitivity in the appendix is valuable.
- The comparison of compouding errors is very useful and shows how FocalPolicy bridges the inter-chunks discontinuities nicely.
- N and H ablations: very good range of values. Ablations in general: excellent experimental design.


Weaknesses:
- The authors should make the demonstration dataset sizes front-and-center (num trajs + their lengths). It is especially important considering the central role of the macro-trajectories of size NH.
- In 5.1 experimental setup, the authors write “we set H = 4 and N = 3”: remind explicitly what H and N refer to there, e.g. “action chunk size” and "number of aggregated chunks”.
- “This maps to a median time of r ≈ 0.982” (Appendix C.1)  this is such an important piece of information that it should appear in the main paper.


Minor:
- L11, 2nd col: “policyto”
- L201, 2nd col: “the L_time supervises only single-chunk”

---

> ### Author Rebuttal · Authors · 2026-03-26
>
> We thank the reviewer for the thorough review of our manuscript and appendix, and for recognizing our contributions. Particularly, we appreciate the positive feedback on the scalability of our approach, the clarity of our presentation, the extensive experimental evaluation, and our technical analysis regarding gradient interaction and spectral sensitivity. We also appreciate the reviewer’s positive assessment of the paper’s overall interest and experimental strength.
> > __W1-3:__ Clarification on dataset scale, parameter definitions, and the significance of sampling time $r$.
>
> __A1-3:__ We sincerely appreciate the reviewer’s guidance on improving the paper's clarity and for highlighting the importance of the parameters $\mu$ and $\sigma$ in relation to the sampling time $r$.
> We will implement the following changes in the revised manuscript:
> 1) In Section 5.1, under "Simulation experiments" and "Real-world experiments," we will include a detailed description of the demonstration dataset scales (number of trajectories and their lengths).
> 2) In the "Implementation Details" of Section 5.1, we will revise the description of $H$ and $N$ to "we set the action chunk size $H = 4$ and the number of aggregated chunks $N = 3$" for better clarity.
> 3) Acknowledging the importance of $\mu$ and $\sigma$ within the LAS module, we will move the discussion regarding their impact on the median sampling time $r$ from the Appendix to Section 4.2.
>
> We also thank the reviewer for pointing out the grammatical errors and typos (e.g., "policyto" and the description of $L_{time}$). We will correct these and conduct a thorough proofreading of the entire paper to ensure professional quality.
>
> > __Q1:__ Clarification of proximal and macro-trajectory lengths relative to task horizon.
>
> __A1:__ Your understanding is exactly correct. We set the proximal length to $H$ and the total prediction length to $NH$, while the actual task execution trajectory is significantly longer than $NH$. To improve readability, we will add a clear explanation regarding the relationship between $N$, $H$, and the task horizon in Section 4.1 when introducing the Foresight Composite Objective. We have provided a detailed clarification as follows:
>
> *In FCO, $L_{time}$ supervises only a single chunk of length $H$, and $L_{freq}$ supervises multiple chunks spanning a length of $L=NH$, where the complete task trajectory is significantly longer than $L$.* (In Section 4.1)
>
> > __Q2:__ Trajectory length settings for baseline comparison in compounding error analysis.
>
> __A2:__ Thank you for pointing this out. For a fair comparison, in the compounding-error experiments shown in Figure 7 and Figure 9, both FocalPolicy and FlowPolicy utilize a consistent trajectory length of 12 steps. We have added a detailed clarification as follows:
>
> *Under the same 12-step prediction horizon, FocalPolicy tracks the expert trajectories more closely and exhibits significantly lower compounding errors compared to FlowPolicy.* (In Section 5.2)

---

> > ### Author Rebuttal · Reviewer_1ToN · 2026-04-03
> >
> > Thank you for the clarifications.
> >
> > The authors have addressed my questions, and I remain comfortable with my original rating.

---

> > > ### Author Response · Authors · 2026-04-04
> > >
> > > We would like to thank you again for taking the time to carefully read our work and for recognizing our contributions. Your guidance on the details of our writing indeed helps improve the readability of the paper.

---

### Official Review · Reviewer_ATUC · 2026-03-18

**Soundness:** 1
**Presentation:** 2
**Significance:** 1
**Originality:** 2
**Overall Recommendation:** 2
**Confidence:** 5

**Summary:**

This paper proposes a visuomotor imitation learning method that aims to reduce discontinuities between action chunks and improve trajectory coherence over time. Its main contributions are a Foresight Composite Objective, which combines short-term time-domain supervision with multi-chunk frequency-domain regularization, and Locally Anchored Sampling, which improves training signal propagation in flow matching. Experiments on simulation and real-world manipulation tasks show improved success.

**Compliance With Llm Reviewing Policy:**

Affirmed.

**Final Justification:**

After reading the authors’ rebuttal and the comments from other reviewers, I still firmly believe that the submission contains a fundamental technical flaw: the method aims to improve cross-chunk consistency through “planning,” yet this approach is fundamentally limited by the lack of memory.

Equally concerning, most of the experiments are conducted on scripted unimodal demonstrations rather than multimodal demonstrations. As a result, the empirical setup is not convincing for evaluating cross-chunk consistency and likely obscures the underlying challenge.

**Key Questions For Authors:**

1. Could the authors clarify why they view planning, rather than the lack of action memory, as the main source of cross-chunk incoherence?

2. Could the authors clarify the main conceptual and empirical advantages of their method over prior refinement-style approaches, such as Reactive Diffusion Policy?

3. More broadly, could the authors explain why their specific design is better suited to improving cross-chunk coherence than the existing methods discussed above?

3. Could the authors comment on why the evaluation does not include more recent benchmarks with human demonstrations, such as RoboMimic, which seem particularly relevant for testing coherence?

**Limitations:**

Yes

**Strengths And Weaknesses:**

Strengths

- Important problem: The paper tackles an interesting and important limitation of chunk-based visuomotor policies—inter-chunk discontinuities in long-horizon manipulation—which is well motivated as a bottleneck for coherent robotic control.
- Well-structured paper: The paper is generally well organized and easy to follow.
- Clear visual presentation: Figures 1–3 make the core intuition and method easy to follow, and the later qualitative figures on real-world tasks and compounding errors help connect the technical ideas.

Weaknesses

- Questionable motivation: The paper argues that cross-chunk coherence mainly requires `plan fine-grained immediate actions and coarse subsequent trajectories`. To my knowledge, this is the core challenge. Recent works have shown that a more fundamental issue is the lack of action memory [1,2]: regardless of how far ahead a policy plans, its future predictions may still become inconsistent with previous chunks.

- Limited novelty: the key idea of `intra-chunk refinement & multi-chunk planning` is not new. Recent work, e.g. Reactive Diffusion Policy [3], has already explored hierarchical refinement-style policy designs and is well recognized by the robot learning community. The paper should clarify why their specific design is more preferable relative to existing methods.

- Unconvincing experimental setup: The simulation benchmarks are outdated (all before 2021), the demonstrations are generated by scripted experts rather than humans (likely limits behavioral diversity and makes the setting less representative of practical chunking challenges), and the baselines do not include any recent methods explicitly designed to address cross-chunk coherence (e.g, [1,4,5]). Overall, it is difficult to assess how much the proposed method advances the current state of the art.

[1] Bidirectional Decoding: Improving Action Chunking via Guided Test-Time Sampling, ICLR 25 \
[2] Action Chunking and Data Augmentation Yield Exponential Improvements in Behavior Cloning for Continuous Spaces, ICLR 26 \
[3] Reactive Diffusion Policy: Slow-Fast Visual-Tactile Policy Learning for Contact-Rich Manipulation, RSS 25 \
[4] Real-Time Execution of Action Chunking Flow Policies, NeurIPS 25 \
[5] ACG: Action Coherence Guidance for Flow-based VLA models, ICRA 26

---

> ### Author Rebuttal · Authors · 2026-03-30
>
> We thank you for recognizing our contributions, noting the structure, presentation and figures in paper.
> > __W1:__ Concerns about the motivation.
>
> __A1:__ We appreciate this comment. We agree that action memory [1, 2] effectively mitigates cross-chunk incoherence. However, conditioning action prediction on past action memory is not the only way to ensure coherence. In our view, anticipating more distant future actions during prediction is equally important, and is also more consistent with human behavioral patterns. Even with action memory, BID [1] still needs to consider subsequent plans.
> > __W2:__ Conceptual and advantages over prior refinement-style approaches.
>
> __A2:__ While both FocalPolicy and RDP [3] follow a “fine-and-coarse” philosophy, they differ fundamentally.
> * To respond to real-time tactile feedback, RDP employs a specific "slow-fast" hierarchy to balance complex trajectory modeling with quick reactive behavior. In contrast, FocalPolicy addresses inter-chunk incoherence by modifying the training objective.
> * Advantages: 1) Generality: As a backbone-agnostic optimization, our method demonstrates strong generalizability, and improves DP3 and FlowPolicy by 2.8% and 7.9% respectively (Table 2). 2) Efficiency: FocalPolicy achieves chunks coherence with a simpler pipeline, without the need for task-specific model designs.
>
> > __W3:__ Concern about the reliability of experiments.
>
> __A3:__ We thank you for the constructive feedback. To address your concerns, we have expanded our evaluations as follows:
> + Broader Benchmarks: We added three tasks from the recent LIBERO (2023) suite. FocalPolicy achieves the highest average success rate, demonstrating the enhanced robustness of FocalPolicy.
> + Diverse Demonstrations: We incorporated four RoboMimic tasks, reporting the average success rate in below table. FocalPolicy exceeds all baselines, proving its effectiveness on complex expert data.
> + Stronger Baselines: We included ACG [5] (integrated into FlowPolicy for a fair comparison). FocalPolicy outperforms ACG across all four benchmarks (A: Adroit, M: MetaWorld, R: RoboMimic, L: LIBERO). Furthermore, FocalPolicy incurs lower mean inference latency (see "Cost" row) as it requires no extra inference modules.
>
> Task|FlowPolicy|FreqPolicy|ACG|Ours
> ---|---|---|---|---
> A: Hammer|0.94|0.94|0.95|1.00
> A: Door|0.53|0.58|0.14|0.63
> A: Pen|0.61|0.53|0.60|0.68
> M: Sweep-Into|0.37|0.19|0.61|0.71
> M: Hand-Insert|0.16|0.16|0.28|0.29
> M: Pick-Place|0.55|0.58|0.55|0.71
> M: Disassemble|0.71|0.85|0.66|0.87
> R: Lift|0.99|0.98|0.99|1.00
> R: Can|0.98|0.90|0.99|1.00
> R: Square|0.74|0.74|0.78|0.82
> R: Transport|0.44|0.38|0.50|0.67
> L: Open-Drawer|0.88|1.00|0.97|1.00
> L: Put-Bowl|0.55|0.70|0.46|0.73
> L: Stack-Bowl|0.54|0.64|0.22|0.45
> Average$\uparrow$|0.64|0.66|0.62|0.75
> Cost (ms)$\downarrow$|5.40|18.90|9.50|5.40
>
> > __Q1:__ Reason for focusing on planning rather than action memory.
>
> __A1:__ As clarified in W&A1, while action memory is valuable for coherence, enhancing a policy’s inherent understanding of trajectory continuity is equally critical. Unlike the "backward-looking" smoothing provided by action memory, FocalPolicy proactively addresses incoherence by ensuring each chunk aligns with a broader future blueprint.
> > __Q2:__ Compare with prior refinement-style approaches.
>
> __A2:__ We detail the conceptual advantages of our method in W&A2. Without the need for task-specific model designs, our method is a plug-and-play approach applicable to diffusion- and flow-based models, improving DP3 and FlowPolicy by 2.8% and 7.9%, respectively (Table 2).
> > __Q3:__ Why FocalPolicy is better suited for improving cross-chunk coherence.
>
> __A3:__ FocalPolicy fundamentally addresses cross-chunk coherence during policy training, offering advantages over the existing methods:
>
> * vs. Inference Modification (BID [1], RTC [4], ACG [5]): By employing FCO during training, we embed coherence directly into the optimization objective. Unlike methods that alter the inference process of pre-trained models, FocalPolicy works within the model itself without adding any computational overhead and sampling latency.
> * vs. Structural Refinement (RDP [3]): Our method serves as a plug-and-play optimization for broad diffusion and flow-based models. Unlike task-specific architectures, FocalPolicy explores a more universal approach for base models to improve action chunk coherence natively.
>
> > __Q4:__ We extend the benchmarks in W&A3.
>
> These discusions and experiments above will be integrated into the revised version.
>
> [1] Bidirectional Decoding: Improving Action Chunking via Guided Test-Time Sampling.
> [2] Action Chunking and Data Augmentation Yield Exponential Improvements in Behavior Cloning for Continuous Spaces.
> [3] Reactive Diffusion Policy: Slow-Fast Visual-Tactile Policy Learning for Contact-Rich Manipulation.
> [4] Real-Time Execution of Action Chunking Flow Policies.
> [5] ACG: Action Coherence Guidance for Flow-based VLA models.

---

> > ### Author Rebuttal · Reviewer_ATUC · 2026-04-04
> >
> > Thank you for the rebuttal. Could you please clarify the input to your policy? Specifically, is your policy conditioned on past action chunks?

---

> > > ### Author Response · Authors · 2026-04-04
> > >
> > > We thank the reviewer for the response. We first clarify that the input to our policy consists of the environmental observations from the current time step $t$ and the previous step $t-1$ (totaling two time steps). This setup does not involve past action chunks (i.e., sequences exceeding two steps). The observations consist of downsampled point clouds and robot proprioception. Specifically, our policy is conditioned on the observations of the current and previous time steps, rather than past action chunks. These settings are consistent with the configurations used in prior policies such as FlowPolicy [1] and DP3 [2]. We will clarify the policy’s inputs and outputs in the revised version to enhance the readability of the paper.
> > >
> > > Please let us know if you have any further questions or suggestions; we are more than happy to address them.
> > >
> > > [1] FlowPolicy: Enabling Fast and Robust 3D Flow-based Policy via Consistency Flow Matching for Robot Manipulation.
> > > [2] 3D Diffusion Policy: Generalizable Visuomotor Policy Learning via Simple 3D Representations.

---

### Decision · Program_Chairs · 2026-04-30

**Decision:**

Accept (regular)

**Comment:**

The paper tackles cross-chunk coherence in diffusion policies via intra-chunk refinement and multi-chunk planning. One reviewer raises major concerns: the motivation is questionable (action memory may be the real issue), novelty is limited (similar hierarchical designs exist, e.g., Reactive Diffusion Policy), and the experimental setup is weak—outdated benchmarks, scripted demonstrations, and no comparison with recent cross-chunk methods. Another reviewer praises the simplicity, strong ablations, spectral analysis, and extensive real-world validation.

Overall, the positive internal quality overcomes the lack of novelty and missing baseline comparisons.